# Weak Supervision Performance Evaluation via Partial Identification

**Felipe Maia Polo**[*]
Department of Statistics
University of Michigan

**Subha Maity**[*]
Department of Statistics and Actuarial Science
University of Waterloo

**Mikhail Yurochkin**
MIT-IBM Watson AI Lab

**Moulinath Banerjee**
Department of Statistics
University of Michigan

**Yuekai Sun**
Department of Statistics
University of Michigan

## Abstract

Programmatic Weak Supervision (PWS) enables supervised model training without direct access to ground truth labels, utilizing weak labels from heuristics, crowd-sourcing, or pre-trained models. However, the absence of ground truth complicates model evaluation, as traditional metrics such as accuracy, precision, and recall cannot be directly calculated. In this work, we present a novel method to address this challenge by framing model evaluation as a partial identification problem and estimating performance bounds using Fréchet bounds. Our approach derives reliable bounds on key metrics without requiring labeled data, overcoming core limitations in current weak supervision evaluation techniques. Through scalable convex optimization, we obtain accurate and computationally efficient bounds for metrics including accuracy, precision, recall, and F1-score, even in high-dimensional settings. This framework offers a robust approach to assessing model quality without ground truth labels, enhancing the practicality of weakly supervised learning for real-world applications.[2]

## 1 Introduction

Programmatic weak supervision (PWS) is a modern learning paradigm that allows practitioners to train their supervised models without the immediate need for ground truth labels $Y$ [50, 48, 47, 49, 58, 64]. In PWS, practitioners first acquire cheap and abundant weak labels $Z$ through heuristics, crowdsourcing, external APIs, and pretrained models, which serve as proxies for $Y$. Then, they fit a *label model*, *i.e.*, a graphical model for $P_{Y,Z}$ [50, 49, 22, 17], which, under appropriate modeling assumptions, can be fitted without requiring $Y$'s. Finally, a predictor $h : \mathcal{X} \to \mathcal{Y}$ is trained using samples $(X_i, Z_i)$'s and a *noise-aware loss* constructed using this fitted label model [50].

One major unsolved issue with the weak supervision approach is that even if we knew $P_{Y,Z}$, evaluation metrics such as accuracy, recall, precision, or $F_1$ cannot be estimated for model validation without any ground truth labels. In fact, these quantities are not identifiable (not uniquely determined) since we only have partial information about the joint distribution $P_{X,Y}$ through the marginals $P_{X,Z}$ and $P_{Y,Z}$. As a consequence, any performance metric based on $h$ cannot be estimated without making extra strong assumptions, *e.g.*, $X \perp\!\!\!\perp Y \mid Z$. Unfortunately, these conditions are unlikely to arise in many situations. A recent work [66] investigated the role and importance of ground truth labels on model evaluation in the weak supervision literature. They determined that, under the current situation,

---

[*]Equal contribution. Corresponding author: Felipe Maia Polo <felipemaiapolo@gmail.com>
[2]Our code can be found on https://github.com/felipemaiapolo/wsbounds

38th Conference on Neural Information Processing Systems (NeurIPS 2024).

the *good performance and applicability of weakly supervised classifiers heavily rely on the presence of at least some high-quality labels, which undermines the purpose of using weak supervision* since models can be directly fine-tuned on those labels and achieve similar performance. Therefore, in this work, we develop new evaluation methods that can be used without any ground truth labels and show that the performance of weakly supervised models can be accurately estimated in many cases, even permitting successful model selection. Our solution relies on partial identification by estimating Fréchet bounds for bounding performance metrics such as accuracy, precision, recall, and $F_1$ score of classifiers trained with weak supervision.

**Fréchet bounds:** Consider a random vector $(X, Y, Z) \in \mathcal{X} \times \mathcal{Y} \times \mathcal{Z}$ is drawn from an unknown distribution $P$. We assume $\mathcal{X} \subset \mathbb{R}^d$ is arbitrary while $\mathcal{Y}$ and $\mathcal{Z}$ are finite. In this work, we develop and analyze the statistical properties of a method for estimating Fréchet bounds [53, 54] of the form

$$L \triangleq \inf_{\pi \in \Pi} \mathbb{E}_{\pi}[g(X, Y, Z)] \text{ and } U \triangleq \sup_{\pi \in \Pi} \mathbb{E}_{\pi}[g(X, Y, Z)] \tag{1.1}$$

when $g$ is a fixed bounded function, with $\Pi$ being the set of distributions $\pi$ for $(X, Y, Z)$ such that the marginal $\pi_{X,Z}$ (resp. $\pi_{Y,Z}$) is identical to the prescribed marginal $P_{X,Z}$ (resp. $P_{Y,Z}$). Our proposed method can efficiently obtain estimates for the bounds by solving convex programs, with the significant advantage that the computational complexity of our algorithm does not scale with the dimensionality of $X$, making it well-suited for applications dealing with high-dimensional data. In previous work, for example, Fréchet bounds were studied in the financial context (*e.g.*, see Rüschendorf [55], Bartl et al. [7]). However, our focus is on applying our methods of estimating Fréchet bounds to the problem of assessing predictors trained using programmatic weak supervision (PWS). For example, the upper and lower bounds for the accuracy of a classifier $h$ can be estimated using our method simply by letting $g(x, y, z) = \mathbb{1}[h(x) = y]$ in (1.1). At a high level, our method replaces $P_{Y,Z}$ with the fitted label model, and $P_{X,Z}$ with its empirical version in the Fréchet bounds in (1.1), and reformulates the problem in terms of a convex optimization problem.

**Contributions:** Our contributions are

1. Developing a practical algorithm for estimating the Fréchet bounds in (1.1). Our algorithm can be summarized as solving convex programs and is scalable to high-dimensional distributions.

2. Quantifying the uncertainty in the computed bounds due to uncertainty in the prescribed marginals by deriving the asymptotic distribution for our estimators.

3. Applying our method to bounding the accuracy, precision, recall, and $F_1$ score of classifiers trained with weak supervision. This enables practitioners to evaluate classifiers in weak supervision settings *without access to ground truth labels*.

## 1.1 Related work

**Weak supervision:** With the emergence of data-hungry models, the lack of properly labeled datasets has become a major bottleneck in the development of supervised models. One approach to overcome this problem is using programmatic weak supervision (PWS) to train predictors in the absence of high-quality labels $Y$ [50, 48, 47, 49, 58, 64]. PWS has shown the potential to solve a variety of tasks in different fields with satisfactory performance. For example, some works have applied weak supervision to named-entity recognition [32, 21, 57], video frame classification [22], bone and breast tumor classification [61]. More recently, Smith et al. [59] proposed a new approach to integrating weak supervision and pre-trained large language models (LLMs). Rather than applying LLMs in the usual zero/few-shot fashion, they treat those large models as weak labelers that can be used through prompting to obtain weak signals instead of using hand-crafted heuristics. Recently, Zhu et al. [66] showed that in many situations, the success of weakly supervised classifiers depends on the availability of ground truth validation samples, undermining the purpose of weak supervision. Then, we develop a new method for model evaluation that does not depend on the availability of any ground truth labels.

A relevant line of research within the realm of weak supervision that is closely related to this work is adversarial learning [4, 5, 42, 41]. Often, adversarial learning aims to learn predictors that perform well in worst-case scenarios. For example, Mazzetto et al. [41] develops a method to learn weakly supervised classifiers in the absence of a good label model. In their work, the authors use a small set of labeled data points to constrain the space of possible data distributions and then find a predictor that performs well in the worst-case scenario. Our work relates to this literature in the sense that we

are interested in the worst and best-case scenarios over a set of distributions. However, we focus on developing an evaluation method instead of another adversarial learning strategy.

**Partial identification:** It is often the case that the distributions of interest cannot be fully observed, which is generally due to missing or noisy data [43, 23]. In cases where practitioners can only observe some aspects of those distributions, *e.g.*, marginal distributions or moments, parameters of interest may not be identifiable without strong assumptions due to ambiguity in the observable data. Partial identification deals with the problem without imposing extra assumptions. This framework allows estimating a set of potential values for the parameters of interest (usually given by non-trivial bounds) and has been frequently considered in many areas such as microeconometrics [38–40, 43], causal inference [25, 23], algorithmic fairness [20, 46]. Our work is most related to Rüschendorf [53, 54, 55], Bartl et al. [7], which study bounds for the uncertainty of a quantity of interest for a joint distribution that is only partially identified through its marginals, *i.e.*, Fréchet bounds. Compared to the aforementioned works, the novelty of our contribution is proposing a convex optimization algorithm that accurately estimates the Fréchet bounds with proven performance guarantees in a setup that is realized in numerous weak-supervision applications.

## 1.2 Notation

We write $\mathbb{E}_Q$ and $\mathrm{Var}_Q$ for the expectation and variance of statistics computed using i.i.d. copies of a random vector $W \sim Q$. Consequently, $\mathbb{P}_Q(A) = \mathbb{E}_Q \mathbb{1}_A$, where $\mathbb{1}_A$ is the indicator of an event $A$. If the distribution is clear by the context, we omit the subscript. If $(a_m)_{m\in\mathbb{N}}$ and $(b_m)_{m\in\mathbb{N}}$ are sequences of scalars, then $a_m = o(b_m)$ is equivalent to $a_m/b_m \to 0$ as $m \to \infty$ and $a_m = b_m + o(1)$ means $a_m - b_m = o(1)$. If $(V^{(m)})_{m\in\mathbb{N}}$ is a sequence of random variables, then (i) $V^{(m)} = o_P(1)$ means that for every $\varepsilon > 0$ we have $\mathbb{P}(|V^{(m)}| > \varepsilon) \to 0$ as $m \to \infty$, (ii) $V^{(m)} = \mathcal{O}_P(1)$ means that for every $\varepsilon > 0$ there exists a $M > 0$ such that $\sup_{m\in\mathbb{N}} \mathbb{P}(|V^{(m)}| > M) < \varepsilon$, (iii) $V^{(m)} = a_m + o_P(1)$ means $V^{(m)} - a_m = o_P(1)$, (iv) $V^{(m)} = o_P(a_m)$ means $V^{(m)}/a_m = o_P(1)$, and (v) $V^{(m)} = \mathcal{O}_P(a_m)$ means $V^{(m)}/a_m = \mathcal{O}_P(1)$.

## 2 Estimating Fréchet bounds

A roadmap to our approach follows. We first reformulate the Fréchet bounds in (1.1) into their dual problems, which we discuss in (2.1). Then, we replace the non-smooth dual problems with their appropriate smooth approximations, as discussed in (2.2). Finally, we propose estimators for the smooth approximations (2.3) and derive their asymptotic distributions in Theorem 2.5.

### 2.1 Dual formulations of the bounds and their approximations

This section presents a result that allows us to efficiently solve the optimization problems in (1.1) by deriving their dual formulations as finite-dimensional convex programs. Before we dive into the result, let us define a family of matrices denoted by

$$\mathcal{A} \triangleq \big\{ a \in \mathbb{R}^{|\mathcal{Y}| \times |\mathcal{Z}|} \; : \; \textstyle\sum_{y\in\mathcal{Y}} a_{yz} = 0 \text{ for every } z \in \mathcal{Z} \big\}.$$

With this definition in place, we introduce the dual formulation in Theorem 2.1.

**Theorem 2.1.** *Let* $g : \mathcal{X} \times \mathcal{Y} \times \mathcal{Z} \to \mathbb{R}$ *be a bounded measurable function. Then,*

$$L = \sup_{a\in\mathcal{A}} \mathbb{E}[f_l(X, Z, a)] \text{ and } U = \inf_{a\in\mathcal{A}} \mathbb{E}[f_u(X, Z, a)] \qquad (2.1)$$

*where*

$$f_l(x, z, a) \triangleq \min_{\bar{y}\in\mathcal{Y}} [g(x, \bar{y}, z) + a_{\bar{y}z}] - \mathbb{E}_{P_{Y|Z}}[a_{Yz}|Z = z]$$

$$f_u(x, z, a) \triangleq \max_{\bar{y}\in\mathcal{Y}} [g(x, \bar{y}, z) + a_{\bar{y}z}] - \mathbb{E}_{P_{Y|Z}}[a_{Yz}|Z = z].$$

*Moreover,* $L$ *and* $U$ *are attained by some optimizers in* $\mathcal{A}$.

Theorem 2.1 remains valid if we maximize/minimize over $\mathbb{R}^{|\mathcal{Y}| \times |\mathcal{Z}|}$ instead of $\mathcal{A}$. However, this is not necessary because the values of $f_l$ and $f_u$ remain identical for the following shifts in $a$: $a_{\cdot z} \leftarrow a_{\cdot z} + b_z$ where $b_z \in \mathbb{R}$. By constraining the set of optimizers to $\mathcal{A}$, we eliminate the possibility of having multiple optimal points. The proof of Theorem 2.1 is placed in Appendix B and is inspired by ideas from Optimal Transport; see Appendix A.

The computation of these bounds entails finding a minimum or maximum over a discrete set, meaning that straightforward application of their empirical versions could result in optimizing non-smooth functions, which is often challenging. To mitigate this, we consider a smooth approximation of the problem that is found to be useful in handling non-smooth optimization problems [3, 6]. We approximate the $\max$ and $\min$ operators with their "soft" counterparts:

$$\text{softmin}\{b_1, \cdots, b_K\} \triangleq -\varepsilon \log[\tfrac{1}{K} \textstyle\sum_k \exp(\tfrac{-b_k}{\varepsilon})], \quad \text{softmax}\{b_1, \cdots, b_K\} \triangleq \varepsilon \log[\tfrac{1}{K} \textstyle\sum_k \exp(\tfrac{b_k}{\varepsilon})],$$

where $\varepsilon > 0$ is a small constant that dictates the level of smoothness. As $\varepsilon$ nears zero, these soft versions of $\max$ and $\min$ converge to their original non-smooth forms. Using these approximations, we reformulate our dual optimization in (2.1) into smooth optimization problems:

$$L_\varepsilon \triangleq \sup_{a \in \mathcal{A}} \mathbb{E}[f_{l,\varepsilon}(X, Z, a)] \text{ and } U_\varepsilon \triangleq \inf_{a \in \mathcal{A}} \mathbb{E}[f_{u,\varepsilon}(X, Z, a)] \tag{2.2}$$

where

$$f_{l,\varepsilon}(x, z, a) \triangleq -\varepsilon \log\left[\tfrac{1}{|\mathcal{Y}|} \sum_{y \in \mathcal{Y}} \exp\left(\tfrac{g(x,y,z)+a_{yz}}{-\varepsilon}\right)\right] - \mathbb{E}_{P_{Y|Z}}[a_{Yz} \mid Z = z]$$

$$f_{u,\varepsilon}(x, z, a) \triangleq \varepsilon \log\left[\tfrac{1}{|\mathcal{Y}|} \sum_{y \in \mathcal{Y}} \exp\left(\tfrac{g(x,y,z)+a_{yz}}{\varepsilon}\right)\right] - \mathbb{E}_{P_{Y|Z}}[a_{Yz} \mid Z = z]$$

and $\varepsilon > 0$ is kept fixed at an appropriate value. As a consequence of Lemma 5 of An et al. [3], we know that $L_\varepsilon$ and $U_\varepsilon$ are no more than $\varepsilon \log |\mathcal{Y}|$ units from $L$ and $U$. Thus, that distance can be regulated by adjusting $\varepsilon$. For example, if we are comfortable with an approximation error of $10^{-2}$ units when $|\mathcal{Y}| = 2$, we will set $\varepsilon = 10^{-2}/\log(2) \approx .014$.

## 2.2 Estimating the bounds

In practice, it is not usually possible to solve the optimization problems in (2.2), because we may not have direct access to the distributions $P_{X,Z}$ and $P_{Y|Z}$. We overcome this problem by assuming that we can estimate the distributions using an available dataset.

To this end, let us assume that we have a sample $\{(X_i, Z_i)\}_{i=1}^n \overset{\text{iid}}{\sim} P_{X,Z}$, and thus we replace the relevant expectations with $P_{X,Z}$ by its empirical version. Additionally, we have a sequence $\{\hat{P}_{Y|Z}^{(m)}, m \in \mathbb{N}\}$ that estimates $P_{Y|Z}$ with greater precision as $m$ increases. Here, $m$ can be viewed as the size of a sample to estimate $P_{Y|Z}$. Although the exact procedure for estimating the conditional distribution is not relevant to this section, we have discussed in our introductory section that this can be estimated using a *label model* [49, 22] in applications with weak supervision or in a variety of other ways for applications beyond weak supervision. Later in this section, we will formalize the precision required for the estimates. To simplify our notation, we omit the superscript $m$ in $\hat{P}_{Y|Z}^{(m)}$, whenever it is convenient to do so.

Thus, the Fréchet bounds are estimated as

$$\hat{L}_\varepsilon = \sup_{a \in \mathcal{A}} \tfrac{1}{n} \textstyle\sum_{i=1}^n \hat{f}_{l,\varepsilon}(X_i, Z_i, a) \text{ and } \hat{U}_\varepsilon = \inf_{a \in \mathcal{A}} \tfrac{1}{n} \textstyle\sum_{i=1}^n \hat{f}_{u,\varepsilon}(X_i, Z_i, a) \tag{2.3}$$

where

$$\hat{f}_{l,\varepsilon}(x, z, a) \triangleq -\varepsilon \log\left[\tfrac{1}{|\mathcal{Y}|} \sum_{y \in \mathcal{Y}} \exp\left(\tfrac{g(x,y,z)+a_{yz}}{-\varepsilon}\right)\right] - \mathbb{E}_{\hat{P}_{Y|Z}}[a_{Yz} \mid Z = z]$$

$$\hat{f}_{u,\varepsilon}(x, z, a) \triangleq \varepsilon \log\left[\tfrac{1}{|\mathcal{Y}|} \sum_{y \in \mathcal{Y}} \exp\left(\tfrac{g(x,y,z)+a_{yz}}{\varepsilon}\right)\right] - \mathbb{E}_{\hat{P}_{Y|Z}}[a_{Yz} \mid Z = z]$$

In our practical implementations we eliminate the constraint that $\sum_y a_{yz} = 0$ for all $z \in \mathcal{Z}$ by adding a penalty term $\sum_{z \in \mathcal{Z}}(\sum_{y \in \mathcal{Y}} a_{yz})^2$ to $\hat{U}_\varepsilon$ (and its negative to $\hat{L}_\varepsilon$) and then solve unconstrained convex programs using the L-BFGS algorithm [33]. Since the penalty term vanishes only when $\sum_y a_{yz} = 0$ for all $z \in \mathcal{Z}$, we guarantee that the optimal solution is in $\mathcal{A}$.

## 2.3 Asymptotic properties of the estimated bounds

In the following, we state the assumptions required for our asymptotic analysis of $\hat{L}_\varepsilon$ and $\hat{U}_\varepsilon$. We start with some regularity assumptions.

**Assumption 2.2.** $L_\varepsilon$ and $U_\varepsilon$ are attained by some optimizers in $\mathcal{A}$ (2.2).

**Assumption 2.3.** *Let $\hat{a}$ represent the optimizer for any problem in* (2.3)*, which is assumed to exist. Suppose $\|\hat{a}\|_\infty = \mathcal{O}_P(1)$ as $m \to \infty$.*

We show in Lemmas C.4, C.5, and C.6 that Assumptions 2.2 and 2.3 can be derived in the binary classification case ($|\mathcal{Y}| = 2$) if $\mathbb{P}(Y = y \mid Z = z)$ is bounded away from both zero and one, *i.e.* $\kappa < \mathbb{P}(Y = y \mid Z = z) < 1 - \kappa$ for some $\kappa > 0$ for every $y \in \mathcal{Y}$ and $z \in \mathcal{Z}$.

In our next assumption, we formalize the degree of precision for the sequence $\{\hat{P}_{Y|Z}^{(m)}, m \in \mathbb{N}\}$ of estimators that we require for desired performances of the bound estimates.

**Assumption 2.4.** *Denote the total variation distance (TV) between probability measures as $d_{\text{TV}}$. For every $z \in \mathcal{Z}$, for some $\lambda > 0$, we have that $d_{\text{TV}}\big(\hat{P}_{Y|Z=z}^{(m)}, P_{Y|Z=z}\big) = \mathcal{O}_P(m^{-\lambda})$.*

From Ratner et al. [49]'s Theorem 2 and a Lipschitz property of the label model[3], we can conclude $\lambda = 1/2$ for a popular label model used in the PWS literature. The asymptotic distributions for the estimated bounds follow.

**Theorem 2.5.** *Assume 2.2, 2.3, and 2.4, and let $n$ be a function of $m$ such that $n \to \infty$ and $n = o(m^{2\lambda})$ when $m \to \infty$. Then, as $m \to \infty$*

$$\sqrt{n}(\hat{L}_\varepsilon - L_\varepsilon) \Rightarrow N(0, \sigma_{l,\varepsilon}^2) \text{ and } \sqrt{n}(\hat{U}_\varepsilon - U_\varepsilon) \Rightarrow N(0, \sigma_{u,\varepsilon}^2)$$

*where $\sigma_{l,\varepsilon}^2 \triangleq \operatorname{Var} f_{l,\varepsilon}(X, Z, a_{l,\varepsilon}^*)$, $\sigma_{u,\varepsilon}^2 \triangleq \operatorname{Var} f_{u,\varepsilon}(X, Z, a_{u,\varepsilon}^*)$, and $a_{l,\varepsilon}^*$ and $a_{u,\varepsilon}^*$ are the unique optimizers to attain $L_\varepsilon$ and $U_\varepsilon$ (2.2).*

Theorem 2.5 tells us that, if the label model is consistent (Assumption 2.4), under some mild regularity conditions (Assumption 2.2 and 2.3), our estimators and will be asymptotically Gaussian with means $L_\varepsilon$ and $U_\varepsilon$ and variances $\sigma_{l,\varepsilon}^2/n$ and $\sigma_{u,\varepsilon}^2/n$. The above theorem requires $m^{2\lambda}$ to grow faster than $n$ implying that, through assumption 2.4, $P_{Y|Z}$ is estimated with a precision greater than the approximation error when we replace $P_{X,Z}$ with $\frac{1}{n} \sum_i \delta_{X_i, Z_i}$. In the case which $\lambda = 1/2$, this condition translates to $n/m \to 0$ as $n \to \infty$. This allows us to derive the asymptotic distribution when combined with classical results from M-estimation (see proof in Appendix C).

**Construction of confidence bounds:** One interesting use of Theorem 2.5 is that we can construct an approximate confidence interval for the estimates of the bounds. For example, an approximate $1 - \gamma$ confidence interval for $L_\varepsilon$ can is constructed as

$$\hat{I} = \left[\hat{L}_\varepsilon - \frac{\tau_\gamma \hat{\sigma}_{l,\varepsilon}}{\sqrt{n}}, \ \hat{L}_\varepsilon + \frac{\tau_\gamma \hat{\sigma}_{l,\varepsilon}}{\sqrt{n}}\right],$$

where $\tau_\gamma = \Phi^{-1}(1 - \gamma/2)$ and $\hat{\sigma}_{l,\varepsilon}$ is the empirical standard deviation of $f_{l,\varepsilon}(X, Z, \cdot)$, substituting the estimate $\hat{a}$ (solution for the problem in 2.3). For such interval, it holds $\mathbb{P}\big(L_\varepsilon \in \hat{I}\big) \approx 1 - \gamma$, *i.e.*, with approximately $1 - \gamma$ confidence we can say that the true $L_\varepsilon$ is in the interval above. An interval for $U_\varepsilon$ can be constructed similarly.

## 3 Evaluation of model performance in weak supervision

In this section, we describe how to use the ideas presented in Section 2 to estimate non-trivial bounds for the evaluation metrics of a weakly supervised classifier $h$ when no high-quality labels are available. In the standard weak supervision setup, only unlabeled data ($X$) is available, but the practitioner can extract weak labels ($Z$) from the available data. More specifically, we assume access to the dataset $\{(X_i, Z_i)\}_{i=1}^m$, i.i.d. with distribution $P_{X,Z}$, used in its entirety to estimate a label model $\hat{P}_{Y|Z}$ [49, 22] and where part of it, *e.g.*, a random subset of size $n$, is used to estimate bounds[4]. To simplify the exposition, we assume the classifier $h$ is fixed[5].

### 3.1 Risk and accuracy

Let $\ell$ be a generic classification loss function. The risk of a classifier $h$ is defined as $R(h) = \mathbb{E}[\ell(h(X), Y)]$, which cannot be promptly estimated in a weak supervision problem, where we do not observe any $Y$. In this situation, we can make use of our bound estimators in Section 2.2, where we set $g(x, y, z) = \ell(h(x), y)$ to obtain bounds for $R(h)$. Furthermore, we can estimate an uncertainty set for the accuracy of the classification simply by letting $g(x, y, z) = \mathbb{1}[h(x) = y]$.

---

[3]Ratner et al. [49] derive that in page 24 of their arXiv paper version.

[4]It is also possible to use distinct datasets for estimating the label model and the bounds as long as the dataset to estimate the label is much bigger. This is our approach in the experiments.

[5]In practice, it can be obtained using all the data not used to estimate bounds.

## 3.2 Precision, recall, and $F_1$ score

For a binary classification problem, where $\mathcal{Y} = \{0, 1\}$, the precision, recall, and $F_1$ score of a classifier $h$ are defined as

$$p \triangleq \mathbb{P}(Y = 1 \mid h(X) = 1) = \frac{\mathbb{P}(h(X)=1, Y=1)}{\mathbb{P}(h(X)=1)}, \quad r \triangleq \mathbb{P}(h(X) = 1 \mid Y = 1) = \frac{\mathbb{P}(h(X)=1, Y=1)}{\mathbb{P}(Y=1)},$$

$$F \triangleq \frac{2}{r^{-1} + p^{-1}} = \frac{2\mathbb{P}(h(X)=1, Y=1)}{\mathbb{P}(h(X)=1) + \mathbb{P}(Y=1)}.$$

The quantities $\mathbb{P}(h(X) = 1)$ and $\mathbb{P}(Y = 1)$ in the above definitions are identified, since the marginals $P_{X,Z}$ and $P_{Y,Z}$ are specified in the Fréchet problem in (1.1). The $\mathbb{P}(h(X) = 1)$ can be estimated from the full dataset $\{(X_i, Z_i)\}_{i=1}^m$ simply using $\hat{\mathbb{P}}(h(X) = 1) \triangleq \frac{1}{m} \sum_{i=1}^m \mathbb{1}[h(X_i) = 1]$. On the other hand, in most weak supervision applications, $\mathbb{P}(Y = 1)$ is assumed to be known from some prior knowledge or can be estimated from an auxiliary dataset, *e.g.*, using the method described in the appendix of Ratner et al. [49]. Estimating or knowing $\mathbb{P}(Y = 1)$ is required to fit the label model [49, 22] in the first place, so it is beyond our scope of discussion. Then, we assume we have an accurate estimate $\hat{\mathbb{P}}(Y = 1)$.

The probability $\mathbb{P}(h(X) = 1, Y = 1)$, which is the final ingredient in the definition of precision, recall, and F1 score is not identifiable as $P_{X,Y}$ is unknown. The uncertainty bounds for this quantity can be estimated using our method simply by letting $g(x, y, z) = \mathbb{1}[h(x) = 1 \text{ and } y = 1]$. Let $\hat{L}_\varepsilon$ and $\hat{U}_\varepsilon$ denote the estimated lower and upper bounds for $\mathbb{P}(h(X) = 1, Y = 1)$ obtained using (2.3). Naturally, the lower bound estimators for precision, recall, and $F_1$ score are

$$\hat{p}_{l,\varepsilon} \triangleq \frac{\hat{L}_\varepsilon}{\hat{\mathbb{P}}(h(X)=1)}, \quad \hat{r}_{l,\varepsilon} \triangleq \frac{\hat{L}_\varepsilon}{\hat{\mathbb{P}}(Y=1)}, \quad \text{and } \hat{F}_{l,\varepsilon} \triangleq \frac{2\hat{L}_\varepsilon}{\hat{\mathbb{P}}(h(X)=1) + \hat{\mathbb{P}}(Y=1)},$$

while the upper bound estimators $\hat{p}_{u,\varepsilon}$, $\hat{r}_{u,\varepsilon}$, and $\hat{F}_{u,\varepsilon}$ are given by substituting $\hat{L}_\varepsilon$ by $\hat{U}_\varepsilon$ above. In the following corollary, we show that the bounds converge asymptotically to normal distributions, which we use for calculating their coverage bounds presented in our applications.

**Corollary 3.1.** *Let $n$ be a function of $m$ such that $n \to \infty$ and $n = o\left(m^{(2\lambda) \wedge 1}\right)$ when $m \to \infty$. Assume the conditions of Theorem 2.5 hold. Then as $m \to \infty$*

○ $\sqrt{n}\left(\hat{p}_{l,\varepsilon} - p_{l,\varepsilon}\right) \Rightarrow N(0, \sigma_{p,l,\varepsilon}^2)$ *with* $p_{l,\varepsilon} = \frac{L_\varepsilon}{\mathbb{P}(h(X)=1)}$, $\sigma_{p,l,\varepsilon}^2 \triangleq \frac{\sigma_{l,\varepsilon}^2}{\mathbb{P}(h(X)=1)^2}$,

○ $\sqrt{n}\left(\hat{r}_{l,\varepsilon} - r_{l,\varepsilon}\right) \Rightarrow N(0, \sigma_{r,l,\varepsilon}^2)$ *with* $r_{l,\varepsilon} = \frac{L_\varepsilon}{\mathbb{P}(Y=1)}$, $\sigma_{r,l,\varepsilon}^2 \triangleq \frac{\sigma_{l,\varepsilon}^2}{\mathbb{P}(Y=1)^2}$,

○ $\sqrt{n}\left(\hat{F}_{l,\varepsilon} - F_{l,\varepsilon}\right) \Rightarrow N(0, \sigma_{F,l,\varepsilon}^2)$ *with* $F_{l,\varepsilon} = \frac{2L_\varepsilon}{\mathbb{P}(h(X)=1)+\mathbb{P}(Y=1)}$ *&* $\sigma_{F,l,\varepsilon}^2 \triangleq \frac{4\sigma_{l,\varepsilon}^2}{[\mathbb{P}(h(X)=1)+\mathbb{P}(Y=1)]^2}$,

*where $L_\varepsilon$, $\sigma_{l,\varepsilon}^2$ are defined in Theorem 2.5. Asymptotic distributions for $\sqrt{n}\left(\hat{p}_{u,\varepsilon} - p_{u,\varepsilon}\right)$, $\sqrt{n}\left(\hat{r}_{u,\varepsilon} - r_{u,\varepsilon}\right)$, and $\sqrt{n}\left(\hat{F}_{u,\varepsilon} - F_{u,\varepsilon}\right)$ are obtained in a similar way by changing $L_\varepsilon$ to $U_\varepsilon$ and $\sigma_{l,\varepsilon}^2$ to $\sigma_{u,\varepsilon}^2$.*

Reiterating our discussion in the final paragraph in Section 2.2, asymptotic distributions are important for constructing confidence intervals for the bounds, which can be done in a similar manner.

# 4  Experiments

All experiments are structured to emulate conditions where high-quality labels are inaccessible during training, validation, and testing phases, and all weakly-supervised classifiers are trained using the noise-aware loss [50]. To fit the label models, we assume $P_Y$ is known (computed using the training set). Unless stated, we use $l_2$-regularized logistic regressors as classifiers, where the regularization strength is determined according to the validation noise-aware loss.

**Wrench datasets:** To carry out realistic experiments within the weak supervision setup and study accuracy/F1 score estimation, we utilize datasets incorporated in Wrench (**W**eak **Supe**rvision **Bench**mark) [63]. This standardized benchmark platform features real-world datasets and pre-generated weak labels for evaluating weak supervision methodologies. Most of Wrench's datasets are designed for classification tasks, encompassing diverse data types such as tabular, text, and image; all contain their pre-computed weak labels. Specifically, we utilize Census [27], YouTube [1], SMS [2], IMDB [37], Yelp [65], AGNews [65], TREC [31], Spouse [12], SemEval [24], CDR [14], ChemProt [29], Commercial [22], Tennis Rally [22], Basketball [22]. For text datasets, we

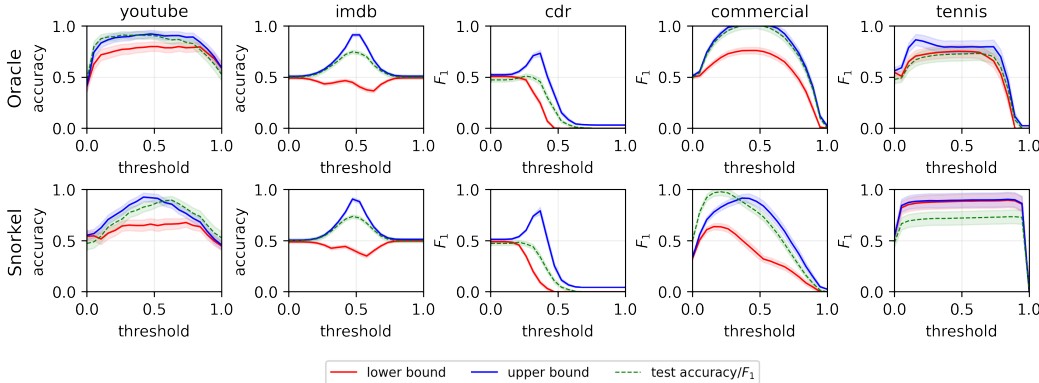

Figure 1: We apply our method to bound test metrics such as accuracy and F1 score (in green) when no true labels are used to estimate performance. In the first row ("Oracle"), we use true labels to estimate the conditional distribution $P_{Y|Z}$, thus approximating a scenario in which the label model is reasonably specified. On the second row ("Snorkel"), we use a label model to estimate $P_{Y|Z}$ without access to any true labels. Despite potential misspecification in Snorkel's label model, it performs comparably to using labels to estimate $P_{Y|Z}$, giving approximate but meaningful bounds.

employ the `paraphrase-MiniLM-L6-v2` model from the *sentence-transformers*[6] library for feature extraction [51]. Features were extracted for the image datasets before their inclusion in Wrench.

**Hate Speech Dataset [15]:** This dataset contains sentence-level annotations for hate speech in English, sourced from posts from white supremacy forums. It encompasses thousands of sentences classified into either `Hate` (1) or `noHate` (0) categories. This dataset provides an ideal ground for examining recall and precision estimation. Social media moderators aim to maximize the filtering of hate posts, *i.e.*, increasing recall, while ensuring that non-hate content is rarely misclassified as offensive, maintaining high precision. Analogously to the Wrench text datasets, we utilize `paraphrase-MiniLM-L6-v2` for feature extraction.

### 4.1 Bounding the performance of weakly supervised classifiers

In this section, we conduct an empirical study using some of the Wrench and Hate Speech datasets to verify the validity and usefulness of our methodology. We compare results for which $P_{Y|Z}$ is estimated using the true labels $Y$ ("Oracle") and those derived using Snorkel's [48, 47] default label model with no hyperparameter tuning and a thousand epochs. Such a comparison facilitates an evaluation of our method's efficacy, especially in cases where the label model could be incorrectly specified. Results for other Wrench datasets and one extra label model (FlyingSquid, [22]) are presented in Appendix F.

In Figure 1, we demonstrate our approaches for bounding test metrics, such as accuracy and F1 score (shown in green), when no true labels are available to estimate performance at various classification thresholds for binary classification tasks on Wrench datasets. In the first row ("Oracle"), true labels are used to estimate the conditional distribution $P_{Y|Z}$, representing a (close to) ideal scenario with a

Table 1: Bounding accuracy in multinomial classification.

| Dataset | Lab. model | Lo. bound | Up. bound | Test acc |
|---------|-----------|-----------|-----------|----------|
| agnews | Oracle | $0.46_{\pm 0.01}$ | $0.95_{\pm 0.01}$ | $0.80_{\pm 0.01}$ |
| | Snorkel | $0.42_{\pm 0.01}$ | $0.9_{\pm 0.01}$ | $0.76_{\pm 0.01}$ |
| semeval | Oracle | $0.54_{\pm 0.04}$ | $0.78_{\pm 0.03}$ | $0.72_{\pm 0.04}$ |
| | Snorkel | $0.36_{\pm 0.03}$ | $0.70_{\pm 0.03}$ | $0.56_{\pm 0.04}$ |

well-specified label model. In the second row ("Snorkel"), however, we use a label model to estimate $P_{Y|Z}$ without relying on any true labels. Despite potential inaccuracies in Snorkel's label model, it achieves results close to those obtained using true labels to estimate $P_{Y|Z}$, yielding approximate but useful bounds. This indicates that even if Snorkel's label model is imperfectly specified, its effectiveness in estimating bounds remains similar to that of the "Oracle" approach, underscoring the value of bounding metrics regardless of label model accuracy. Delving deeper into Figure 1, results for "youtube", "commercial", and "tennis" highlight that our uncertainty about out-of-sample performance is small, even without labeled samples. However, there is a noticeable increase in uncertainty for "imdb" and "cdr", making weakly supervised models deployment riskier without

---

[6]Accessible at https://huggingface.co/sentence-transformers/paraphrase-MiniLM-L6-v2.

additional validation. Yet, the bounds retain their informative nature. For instance, for those willing to accept the risk, the "imdb" classifier's ideal threshold stands at .5. This is deduced from the flat worst-case and peaking best-case accuracy at this threshold. Table 1 presents some results for "agnews" (4 classes) and "semeval" (9 classes). From Table 1, we can see that both "Oracle" and "Snorkel" approaches produce valid bounds.

Now, we present bounds on the classifiers' precision and recall across different classification thresholds for the hate speech dataset. This dataset did not provide weak labels, so we needed to generate them. We employed four distinct weak labelers. The initial weak labeler functions are based on keywords and terms. Should words or phrases match those identified as hate speech in the lexicon created by Davidson et al. [13], we categorize the sentence as 1; if not, it's designated 0. The second weak labeler is based on TextBlob's sentiment analyzer [36]: a negative text polarity results in a 1 classification, while other cases are labeled 0. Our final pair of weak labelers are language

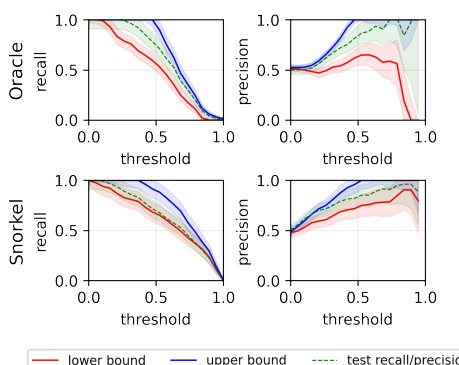

Figure 2: Precision and recall bounds for hate speech detection.

models, specifically BERT [16] and RoBERTa [34], that have undergone fine-tuning for detecting toxic language or hate speech [35, 28]. Figure 2 presents both recall and precision bounds and test estimates for the weakly-supervised hate speech classifier. Mirroring observations from Figure 1, Snorkel's standard label model gives valuable bounds analogous to scenarios where we employ labels to estimate $P_{Y|Z}$. If used by practitioners, Figure 2 could help trade-off recall and precision by choosing an appropriate classification threshold in the absence of high-quality labels.

## 4.2 Choosing a set of weak labels

In this experiment, we examine how our approach performs under the influence of highly informative weak labels as opposed to scenarios with less informative weak labels. Using the YouTube dataset provided by Wrench, we attempt to classify YouTube comments into categories of SPAM or HAM, leveraging Snorkel to estimate $P_{Y|Z}$. Inspired by Smith et al. [59], we craft three few-shot weak labelers by prompting[7] the large language model (LLM) Llama-2-13b-chat-hf [60]. For each dataset entry, we pose three distinct queries to the LLM. Initially, we inquire if the comment is SPAM or HAM. Next, we provide clear definitions of SPAM and HAM, then seek the classification from LLM. In the third prompt, leveraging in-context learning ideas [17], we provide five representative comments labeled as SPAM/HAM prior to requesting the LLM's verdict on the comment in question. In cases where LLM's response diverges from SPAM or HAM, we interpret it as LLM's abstention.

After obtaining this triad of weak labels, we analyze two situations. Initially, we integrate the top five[8] weak labels ("high-quality" labels) from Wrench. In the subsequent scenario, we synthetically generate weak labels ("low-quality" labels) that do not correlate with $Y$. The first plot in Figure 3 depicts the bounds of our classifier based solely on weak few-shot labels, which unfortunately do not provide substantial insights. Enhancing the bounds requires the inclusion of additional weak labels. Yet, as indicated by the subsequent pair of plots, it becomes evident that only the incorporation of "high-quality" weak labels results in significant shrinkage and upward shift of the bounds. As confirmed by the test accuracy, if a practitioner had used our method to select the set of weak labels, that would have led to a significant boost in performance.

## 4.3 Model selection strategies using the Fréchet bounds

In Sections 4.1 and 4.2, we implicitly touched on the topic of model selection when discussing the classification threshold and weak label selection. Here, we explicitly discuss the use of our Fréchet bounds for model selection purposes. Consider a set of possible models $\mathcal{H} \triangleq \{h_1, \cdots, h_K\}$ from which we wish to find the best model according to a specific metric, $e.g.$, accuracy, or F1 score. We consider three approaches for model selection using the Fréchet bounds: choosing the model with the best possible (i) lower bound, (ii) upper bound, and (iii) average of lower and upper bounds on the metric of interest. Strategy (i) works well for the worst-case scenario and can be seen as the

---

[7]More details regarding the prompts can be found in Appendix G.1.

[8]The most informative weak labels are determined based on their alignment with the true label.

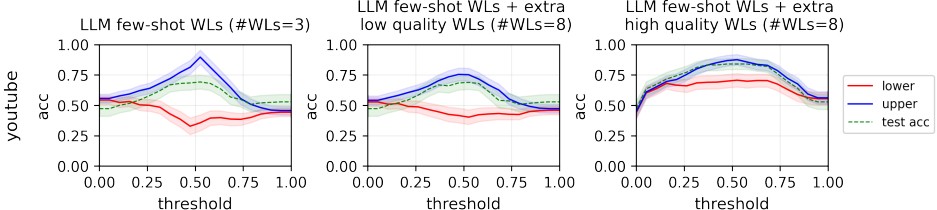

Figure 3: Performance bounds for classifiers on the YouTube dataset, initially relying solely on few-shot weak labels obtained via prompts to the LLM `Llama-2-13b-chat-hf`. The progression of plots illustrates the comparative impact of integrating "high-quality" labels from Wrench versus synthetically generated "low-quality" labels. Evidently, the addition of "high-quality" labels significantly enhances the bounds, underscoring their superior utility over "low-quality" labels for optimal classification of `SPAM` and `HAM` comments.

distributionally robust optimization (DRO) [9] solution when the uncertainty set is given by $\Pi$ in (1.1), while (ii) is suitable for an optimistic scenario, and (iii) is suggested when one wants to balance between the worst- and best-case scenarios. Please check Appendix E for more details.

In this experiment, we select multilayer-perceptrons (MLPs). The considered MLPs have one hidden layer with a possible number of neurons in $\{50, 100\}$. Training is carried out with Adam [26], with possible learning rates in $\{.1, .001\}$ and weight decay ($l_2$ regularization parameter) in $\{.1, .001\}$. For those datasets that use the F1 score as the evaluation metric, we also tune the classification threshold in $\{.2, .4, .5, .6, .8\}$ (otherwise, they return the most probable class as a prediction). In total, $\mathcal{H}$ is composed of 8 trained models when evaluating accuracy and 40 models when evaluating the F1 score. We also consider directly using the label model (Snorkel [47]) to select models. For example, when the metric considered is accuracy, *i.e.*, we use select the model $\arg\max_{h_k \in \mathcal{H}} \frac{1}{n} \sum_{i=1}^{n} \mathbb{E}_{\hat{P}_{Y|Z}} \mathbb{1}[h_k(X) = Y \mid Z = Z_i]$, which is a natural choice when $X \perp\!\!\!\perp Y \mid Z$. As baselines, we consider having a few labeled samples.

In Table 2, we report a subset of our results (please check Appendix E for the full set of results). In this table, we report the average test scores of the chosen models over 10 repetitions for different random seeds (standard deviation re-

Table 2: Performance of selected models

| | metric | Lower bound | Bounds avg | Label model | Labeled ($n = 100$) |
|---|---|---|---|---|---|
| agnews | acc | $0.77_{\pm 0.00}$ | $0.78_{\pm 0.00}$ | $0.77_{\pm 0.00}$ | $0.77_{\pm 0.00}$ |
| imdb | acc | $0.72_{\pm 0.00}$ | $0.73_{\pm 0.00}$ | $0.73_{\pm 0.00}$ | $0.72_{\pm 0.01}$ |
| yelp | acc | $0.81_{\pm 0.00}$ | $0.81_{\pm 0.00}$ | $0.81_{\pm 0.00}$ | $0.82_{\pm 0.01}$ |
| tennis | F1 | $0.76_{\pm 0.01}$ | $0.76_{\pm 0.01}$ | $0.75_{\pm 0.01}$ | $0.71_{\pm 0.02}$ |
| commercial | F1 | $0.96_{\pm 0.00}$ | $0.96_{\pm 0.00}$ | $0.96_{\pm 0.00}$ | $0.91_{\pm 0.01}$ |

port as subscript). We can extract some lessons from the table. First, using metrics derived from the Fréchet bounds is most useful when our uncertainty about the model performance is low, *e.g.*, "'commercial" and "tennis" in Figure 1. In those cases, using our metrics for model selection gives better results even when compared to a labeled validation set of size $n = 100$. Moreover, once the practitioner knows that the uncertainty is low, using the label model approach also does well.

## 5 Discussion

### 5.1 Extensions

An extension we do not address in the main text is the evaluation of end-to-end weak supervision methods [62, 56, 52], where the separation between the label model and the final predictor is less clear than in our primary setting. Our approach remains compatible with these methods as long as we can fit a label model (*e.g.*, Snorkel) separately and utilize it solely for the evaluation step. Another possible extension is the application of the ideas presented in this work in different fields of machine learning or statistics. One could consider applying our ideas to the problem of "statistical matching" (SM) [18, 10, 19, 30, 11], for example. The classic formulation of SM involves observing two distinct datasets that contain replications of $(X, Z)$ and $(Y, Z)$, but the triplet $(X, Y, Z)$ is never observed. The primary goal is to make inferences about the relationship between $X$ and $Y$. For instance, if our focus is on bounding $\mathbb{P}((X, Y) \in B) = \mathbb{E}[\mathbb{1}_B(X, Y)]$ for a certain event $B$, we could define $g(x, y, z) = \mathbb{1}_B(x, y)$ and apply our method.

## 5.2 Limitations

We discuss some limitations of our methods. Firstly, our method and theoretical results are only applicable to cases where $\mathcal{Y}$ and $\mathcal{Z}$ are finite sets, such as in classification problems. Extending the dual formulation in Theorem 2.1 to general $\mathcal{Y}$ and $\mathcal{Z}$ is possible but would require optimizing over function spaces, which is computationally and theoretically challenging. Additionally, if $|\mathcal{Z}|$ is large, convergence may be slow, necessitating a large unlabeled dataset for accurate bounds. Using a smaller, curated set of weak labels, may be more effective for bounds estimation and performance. We end this subsection with two other limitations related to misspecification in the label model and informativeness of the bounds. The proofs of the results introduced in this section are placed in Appendix D.

**Label model misspecification:** In our asymptotic results we assumed that the label models are well-specified, *i.e.*, the estimates $\{\hat{P}_{Y|Z}^{(m)}, m \in \mathbb{N}\}$ converge to the true label model $P_{Y|Z}$ at $m \to \infty$. To understand the qualities of our bound when this assumption is violated, we introduce the misspecification: $\hat{P}_{Y|Z}^{(m)} \to Q_{Y|Z}$ and $Q_{Y|Z} \neq P_{Y|Z}$. In our investigation on the misspecification of the label model, we control the level of misspecification as $d_{\mathrm{TV}}(Q_{Y|Z=z}, P_{Y|Z=z}) \leq \delta$ and then study the subsequent errors in our Fréchet bounds. The following theorem formalizes the result.

**Theorem 5.1.** *Recall from equation* (2.2) *that $L_\epsilon$ and $U_\epsilon$ are the smoothened upper and lower Fréchet bounds with the true $P_{Y|Z=z}$. Additionally, let us define similar $\check{L}_\epsilon$ and $\check{U}_\epsilon$ bounds, but with a misspecified $Q_{Y|Z=z}$, i.e.*

$$\check{L}_\epsilon \triangleq \sup_{a \in \mathcal{A}} \mathbb{E}[\check{f}_{l,\varepsilon}(X, Z, a)] \quad and \quad \check{U}_\epsilon \triangleq \inf_{a \in \mathcal{A}} \mathbb{E}[\check{f}_{u,\varepsilon}(X, Z, a)],$$

$$\check{f}_{l,\varepsilon}(x, z, a) \triangleq -\varepsilon \log \left[ \frac{1}{|\mathcal{Y}|} \sum_{y \in \mathcal{Y}} \exp \left( \frac{g(x,y,z) + a_{yz}}{-\varepsilon} \right) \right] - \mathbb{E}_{Q_{Y|Z}}[a_{Yz} \mid Z = z] \qquad (5.1)$$

$$\check{f}_{u,\varepsilon}(x, z, a) \triangleq \varepsilon \log \left[ \frac{1}{|\mathcal{Y}|} \sum_{y \in \mathcal{Y}} \exp \left( \frac{g(x,y,z) + a_{yz}}{\varepsilon} \right) \right] - \mathbb{E}_{Q_{Y|Z}}[a_{Yz} \mid Z = z]$$

*Assume $Q_{Y|Z}$ is in a set of conditional distributions such that the optimizers for* (5.1)*, which are assumed to exist, are uniformly bounded. If $d_{\mathrm{TV}}(Q_{Y|Z=z}, P_{Y|Z=z}) \leq \delta$, then for some $C > 0$ which is independent of $\delta > 0$, we have*

$$\max\left(|\check{L}_\epsilon - L_\epsilon|, |\check{U}_\epsilon - U_\epsilon|\right) \leq C\delta. \qquad (5.2)$$

The above theorem reveals how misspecification translates to the errors in subsequent Fréchet bounds. In an ideal scenario, with access to an $\{(Y_i, Z_i)\}_{i=1}^m$ sample we can consistently estimate $P_{Y|Z}$, leading to $\delta = 0$. In situations when this $\{(Y_i, Z_i)\}_{i=1}^m$ sample is not accessible and we have to rely on a label model, we require the misspecification in this model to be small.

**Informativeness of the bounds:** The bounds $L$ and $U$ are especially useful when their difference $U - L$ is small because in that case, we obtain a tight bound for $\mathbb{E}[g(X, Y, Z)]$ and narrow it down with high precision even if the joint random vector $(X, Y, Z)$ is never observed. But when is this bound small? In the next theorem, we provide an upper bound on this difference.

**Theorem 5.2.** *Let $L$ and $U$ be defined as in equation* (1.1)*. Then*

$$U - L \leq \sqrt{8 \|g\|_\infty^2 \min\{H(X \mid Z), H(Y \mid Z)\}}$$

*where $H(X \mid Z)$ (resp. $H(Y \mid Z)$) denotes the conditional entropy of $X$ (resp. $Y$) given $Z$.*

To understand the result better, recall our setting: we do not observe the joint distribution $P_{X,Y,Z}$ and only observe the marginals $P_{Y,Z}$ and $P_{X,Z}$. The only way we can infer about the joint distribution is by connecting these two marginals through $Z$. So, readers can guess that it is more favorable when $Z$ is informative for either $X$ or $Y$. For example, take the extreme case when $Y = h(Z)$ for a function $h : \mathcal{Y} \to \mathcal{Z}$. In this case the $P_{X,Y,Z} = P_{X,g(Z),Z}$ is precisely known from the $P_{X,Z}$ and we can exactly pinpoint the $\mathbb{E}[g(X, Y, Z)]$ as $\mathbb{E}[g(X, h(Z), Z)]$. In this case, $H(Y \mid Z) = 0$, leading to $U - L = 0$, *i.e.*, its Fréchet bounds can precisely pinpoint it as well; ideally at least one of the $H(X \mid Z)$ and $H(Y \mid Z)$ is small.

## 6 Acknowledgements

This paper is based upon work supported by the National Science Foundation (NSF) under grants no. 1916271, 2027737, 2113373, and 2113364.

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

## A Connection to optimal transport

The optimizations in the Frechet bounds (1.1) can be connected to an optimization problem [45]. We only explain this connection for the lower bound, but the connection to the upper bound is quite similar. The optimization lower bound is

$$\inf_{\substack{\pi_{X,Z}=P_{X,Z} \\ \pi_{Y|Z}=P_{Y|Z}}} \mathbb{E}_\pi[g(X,Y,Z)] =$$

$$= \inf_{\substack{\pi_{X,Z}=P_{X,Z} \\ \pi_{Y|Z}=P_{Y|Z}}} \sum_z \mathbb{P}(Z=z)\mathbb{E}_\pi[g(X,Y,Z) \mid Z=z]$$

$$= \sum_z \mathbb{P}(Z=z) \left\{ \inf_{\substack{\pi_{X|Z=z}=P_{X|Z=z} \\ \pi_{Y|Z=z}=P_{Y|Z=z}}} \mathbb{E}_{\pi_{X,Y|Z=z}}[g(X,Y,z)] \right\}$$

where we notice that the inner minimization is an optimal transport problem between the probability distributions $P_{X|Z=z}$ and $P_{Y|Z=z}$ with the cost function $d_z(x,y) = g(x,y,z)$.

# B   A proof for the duality result

*Proof of Theorem 2.1.* We start proving the result for $L$. See that

$$L = \inf_{\pi \in \Pi} \mathbb{E}_\pi[g(X, Y, Z)]$$

$$= \inf_{\{\pi_z \in \Pi_z\}_{z \in \mathcal{Z}}} \sum_{z \in \mathcal{Z}} \mathbb{P}(Z = z) \cdot \mathbb{E}_{\pi_z}[g(X, Y, Z) \mid Z = z]$$

$$= \sum_{z \in \mathcal{Z}} \mathbb{P}(Z = z) \cdot \inf_{\pi_z \in \Pi_z} \mathbb{E}_{\pi_z}[g(X, Y, Z) \mid Z = z]$$

with $\Pi_z$, is defined as

$$\Pi_z \triangleq \{\pi_z \in \Delta(\mathcal{X} \times \mathcal{Y}) : \pi_z \circ \rho_X^{-1} = P_{X|Z=z} \text{ and } \pi_z \circ \rho_Y^{-1} = P_{Y|Z=z}\}$$

That is, for each $z \in \mathcal{Z}$, $\Pi_z$ represents the set of couplings such that marginals are given by $P_{X|Z=z}$ and $P_{Y|Z=z}$. We can represent the problem in this way since the marginal distribution of $Z$ is fixed and, given that distribution, $\{\Pi_z\}$ specifies the same set of distributions as $\Pi$.

Realize that we have broken down our initial maximization problem in $|\mathcal{Z}|$ smaller minimization problems. Each of those minimization problems can be treated as an optimal transportation problem. Consequently, by Beiglböck and Schachermayer [8, Theorem 1], for each $z \in \mathcal{Z}$, we get the following duality result

$$\inf_{\pi_z \in \Pi_z} \mathbb{E}_{\pi_z}[g(X, Y, Z) \mid Z = z] = \sup_{(\beta_z, \alpha_z) \in \Psi_z} \mathbb{E}[\beta_z(X) + \alpha_z(Y) \mid Z = z]$$

with

$$\Psi_z \triangleq \left\{ (\beta_z, \alpha_z) : \begin{array}{c} \beta_z : X \to [-\infty, \infty), \alpha_z : Y \to [-\infty, \infty) \\ \mathbb{E}[|\beta_z(X)| \mid Z = z] < \infty, \mathbb{E}[|\alpha_z(Y)| \mid Z = z] < \infty \\ \beta_z(x) + \alpha_z(y) \le g(x, y, z) \text{ for all } (x, y) \in X \times Y \end{array} \right\}$$

Moreover, partially optimizing on $\beta_z(x)$, we can set $\beta_z^*(x) = \min_{y \in \mathcal{Y}}[g(x, y, z) - \alpha_z(y)]$ and then

$$\inf_{\pi_z \in \Pi_z} \mathbb{E}_{\pi_z}[g(X, Y, Z) \mid Z = z] =$$

$$= \sup_{\alpha_z} \mathbb{E}\left[\min_{y \in \mathcal{Y}}[g(X, y, Z) + \alpha_z(y)] - \alpha_z(Y) \mid Z = z\right]$$

where $\alpha_z$ is a simple function taking values in the real line.

Consequently,

$$L = \sum_{z \in \mathcal{Z}} \mathbb{P}(Z = z) \cdot \sup_{\alpha_z} \mathbb{E}\left[\min_{y \in \mathcal{Y}}[g(X, y, Z) + \alpha_z(y)] - \alpha_z(Y) \mid Z = z\right]$$

$$= \sup_{\{\alpha_z\}_{z \in \mathcal{Z}}} \sum_{z \in \mathcal{Z}} \mathbb{P}(Z = z) \cdot \mathbb{E}\left[\min_{y \in \mathcal{Y}}[g(X, y, Z) + \alpha_z(y)] - \alpha_z(Y) \mid Z = z\right]$$

$$= \sup_{\{\alpha_z\}_{z \in \mathcal{Z}}} \mathbb{E}\left[\min_{y \in \mathcal{Y}}[g(X, y, Z) + \alpha_Z(y)] - \alpha_Z(Y)\right]$$

Because each $\alpha_z$ is a function assuming at most $|\mathcal{Y}|$ values and we have $|\mathcal{Z}|$ functions (one for each value of $z$), we can equivalently solve an optimization problem on $\mathbb{R}^{|\mathcal{Y}| \times |\mathcal{Z}|}$. Adjusting the notation,

$$L = \sup_{a \in \mathbb{R}^{|\mathcal{Y}| \times |\mathcal{Z}|}} \mathbb{E}\left[\min_{\bar{y} \in \mathcal{Y}}[g(X, \bar{y}, Z) + a_{\bar{y}Z}]\right] - \mathbb{E}[a_{YZ}]$$

From Beiglböck and Schachermayer [8, Theorem 2], we know that the maximum is attained by some $a^* \in \mathbb{R}^{|\mathcal{Y}| \times |\mathcal{Z}|}$. To show that there is a maximizer in $\mathcal{A}$, we need to update the solution

$$a_{\cdot z}^* \leftarrow a_{\cdot z}^* - \sum_{y \in \mathcal{Y}} a_{yz}^* \quad \text{(shift sub-vector by a constant)} \tag{B.1}$$

for every $z \in \mathcal{Z}$. The objective function is not affected by such translations.

To prove the result for $U$, first realize that because $g$ is bounded, with no loss of generality, we can assume its range is a subset of $[0, 1]$. Define $c(x, y, z) = 1 - g(x, y, z)$ and see that

$$U = \sup_{\pi \in \Pi} \mathbb{E}_\pi[1 - c(X, Y, Z)] = 1 - \inf_{\pi \in \Pi} \mathbb{E}_\pi[c(X, Y, Z)]$$

Proceeding as before, we can obtain the final result by finding the dual formulation for $\inf_{\pi \in \Pi} \mathbb{E}_\pi[c(X, Y, Z)]$. $\qquad \square$

# C   Proofs for the estimation results

**We will analyze the estimator for $U$ (results for the estimator of $L$ can be obtained analogously).**

For the next results, we define

$$\tilde{f}_{u,\varepsilon}(x,z,a) \triangleq f_{u,\varepsilon}(x,z,a) + \sum_{z'\in\mathcal{Z}}\left(\sum_{y\in\mathcal{Y}} a_{yz'}\right)^2.$$

*Proof of Theorem 2.5.* By Assumption 2.2 and Lemmas C.1 and C.2, we can guarantee that $\mathbb{E}[\tilde{f}_{u,\varepsilon}(X,Z,a)]$ is minimized by a unique $a^*_{u,\varepsilon}$ (equalling it to $U_\varepsilon$) and that $\nabla^2_a\mathbb{E}[\tilde{f}_{u,\varepsilon}(X,Z,a^*_{u,\varepsilon})]$ is positive definite. Also, from the proof of Lemma C.2, we can see that $\tilde{f}_{u,\varepsilon}$ is convex in $a$ (because its Hessian is positive semidefinite). It is also true that the second moment of $\tilde{f}_{u,\varepsilon}(X,Z,a)$ is well defined (exists and finite) for each $a$ since $g$ is bounded. Define

$$\tilde{U}_\varepsilon \triangleq \inf_{a\in\mathbb{R}^{|\mathcal{Y}|\times|\mathcal{Z}|}} \frac{1}{n}\sum_{i=1}^n \tilde{f}_{u,\varepsilon}(X_i,Z_i,a)$$

and let $\tilde{a}_\varepsilon$ denote a value that attains that minimum; from Niemiro [44, Theorem 4] and the conditions discussed above, we know that $\sqrt{n}(\tilde{a}_\varepsilon - a^*_{u,\varepsilon}) = \mathcal{O}_P(1)$. The existence of $\tilde{a}_\varepsilon$ is discussed by Niemiro [44]. Then

$$\sqrt{n}(\tilde{U}_\varepsilon - U_\varepsilon)$$
$$= \frac{1}{\sqrt{n}}\left(\sum_i \tilde{f}_{u,\varepsilon}(X_i,Z_i,\tilde{a}_\varepsilon) - \sum_i \tilde{f}_{u,\varepsilon}(X_i,Z_i,a^*_{u,\varepsilon})\right) + \sqrt{n}\left(\frac{1}{n}\sum_i \tilde{f}_{u,\varepsilon}(X_i,Z_i,a^*_{u,\varepsilon}) - U_\varepsilon\right)$$
$$= \frac{1}{\sqrt{n}}\left([\sqrt{n}(\tilde{a}_\varepsilon - a^*_{u,\varepsilon})]^\top[\frac{1}{n}\sum_i \nabla^2_a\tilde{f}_{u,\varepsilon}(X_i,Z_i,\bar{a})][\sqrt{n}(\tilde{a}_\varepsilon - a^*_{u,\varepsilon})]\right) +$$
$$+ \sqrt{n}\left(\frac{1}{n}\sum_i \tilde{f}_{u,\varepsilon}(X_i,Z_i,a^*_{u,\varepsilon}) - U_\varepsilon\right) + o_P(1)$$

(C.1)

where the first term is obtained by a second-order Taylor expansion of the summing functions around $\tilde{a}_\varepsilon$ ($\bar{a}$ is some random vector). Also, from the standard central limit theorem, we know that

$$\sqrt{n}\left(\frac{1}{n}\sum_i \tilde{f}_{u,\varepsilon}(X_i,Z_i,a^*_{u,\varepsilon}) - U_\varepsilon\right) \Rightarrow N(0,\operatorname{Var}\tilde{f}_{u,\varepsilon}(X,Z,a^*_{u,\varepsilon}))$$

Given that $\sqrt{n}(\tilde{a}_\varepsilon - a^*_{u,\varepsilon}) = \mathcal{O}_P(1)$ and that the Hessian has bounded entries (then $\mathcal{O}_P(1)$ as well), the first term in C.1 is $o_P(1)$. Because $a^*_{u,\varepsilon} \in \mathcal{A}$, we have that $f_{u,\varepsilon}(X,Z,a^*_{u,\varepsilon}) = \tilde{f}_{u,\varepsilon}(X,Z,a^*_{u,\varepsilon})$ and then

$$\sqrt{n}(\tilde{U}_\varepsilon - U_\varepsilon) \Rightarrow N(0,\operatorname{Var} f_{u,\varepsilon}(X,Z,a^*_{u,\varepsilon}))$$

by Slutsky's theorem. Since

$$\sqrt{n}(\hat{U}_\varepsilon - U) = \sqrt{n}(\hat{U}_\varepsilon - \tilde{U}_\varepsilon) + \sqrt{n}(\tilde{U}_\varepsilon - U_\varepsilon),$$

if we can show that

$$\sqrt{n}(\tilde{U}_\varepsilon - \hat{U}_\varepsilon) = o_P(1)$$

we are done.

Let $\hat{a}$ be solution for the problem in 2.3 and see that

$$|\tilde{U}_\varepsilon - \hat{U}_\varepsilon| =$$

$$= \left| \frac{1}{n} \sum_{i=1}^n \tilde{f}_{u,\varepsilon}(X_i, Z_i, \tilde{a}_\varepsilon) - \frac{1}{n} \sum_{i=1}^n \varepsilon \log \left[ \frac{1}{|\mathcal{Y}|} \sum_{y \in \mathcal{Y}} \exp \left( \frac{g(X_i, y, Z_i) + \hat{a}_{yZ_i}}{\varepsilon} \right) \right] + \mathbb{E}_{\hat{P}_{Y|Z}}[\hat{a}_{YZ} \mid Z = Z_i] \right|$$

$$\leq \left| \frac{1}{n} \sum_{i=1}^n \tilde{f}_{u,\varepsilon}(X_i, Z_i, \tilde{a}_\varepsilon) - \frac{1}{n} \sum_{i=1}^n \tilde{f}_{u,\varepsilon}(X_i, Z_i, \hat{a}) \right| +$$

$$+ \left| \frac{1}{n} \sum_{i=1}^n \tilde{f}_{u,\varepsilon}(X_i, Z_i, \hat{a}) - \frac{1}{n} \sum_{i=1}^n \varepsilon \log \left[ \frac{1}{|\mathcal{Y}|} \sum_{y \in \mathcal{Y}} \exp \left( \frac{g(X_i, y, Z_i) + \hat{a}_{yZ_i}}{\varepsilon} \right) \right] + \mathbb{E}_{\hat{P}_{Y|Z}}[\hat{a}_{YZ} \mid Z = Z_i] \right|$$

$$= \left( \frac{1}{n} \sum_{i=1}^n \tilde{f}_{u,\varepsilon}(X_i, Z_i, \hat{a}) - \frac{1}{n} \sum_{i=1}^n \tilde{f}_{u,\varepsilon}(X_i, Z_i, \tilde{a}_\varepsilon) \right) +$$

$$+ \left| \frac{1}{n} \sum_{i=1}^n \sum_y [\mathbb{P}(Y = y \mid Z = Z_i) - \hat{\mathbb{P}}(Y = y \mid Z = Z_i)] \hat{a}_{yZ_i} \right|$$

$$\leq \left( \frac{1}{n} \sum_{i=1}^n \tilde{f}_{u,\varepsilon}(X_i, Z_i, \hat{a}) - \frac{1}{n} \sum_{i=1}^n \varepsilon \log \left[ \frac{1}{|\mathcal{Y}|} \sum_{y \in \mathcal{Y}} \exp \left( \frac{g(X_i, y, Z_i) + \hat{a}_{yZ_i}}{\varepsilon} \right) \right] + \mathbb{E}_{\hat{P}_{Y|Z}}[\hat{a}_{YZ} \mid Z = Z_i] \right)$$

$$+ \left( \frac{1}{n} \sum_{i=1}^n \varepsilon \log \left[ \frac{1}{|\mathcal{Y}|} \sum_{y \in \mathcal{Y}} \exp \left( \frac{g(X_i, y, Z_i) + \tilde{a}_{\varepsilon y Z_i}}{\varepsilon} \right) \right] - \mathbb{E}_{\hat{P}_{Y|Z}}[\tilde{a}_{\varepsilon YZ} \mid Z = Z_i] - \frac{1}{n} \sum_{i=1}^n \tilde{f}_{u,\varepsilon}(X_i, Z_i, \tilde{a}_\varepsilon) \right)$$

$$+ \left| \frac{1}{n} \sum_{i=1}^n \sum_y [\mathbb{P}(Y = y \mid Z = Z_i) - \hat{\mathbb{P}}(Y = y \mid Z = Z_i)] \hat{a}_{yZ_i} \right|$$

$$= \left( \frac{1}{n} \sum_{i=1}^n \sum_y [\hat{\mathbb{P}}(Y = y \mid Z = Z_i) - \mathbb{P}(Y = y \mid Z = Z_i)] \hat{a}_{yZ_i} \right)$$

$$+ \left( \frac{1}{n} \sum_{i=1}^n \sum_y [\mathbb{P}(Y = y \mid Z = Z_i) - \hat{\mathbb{P}}(Y = y \mid Z = Z_i)] \tilde{a}_{\varepsilon y Z_i} \right)$$

$$+ \left| \frac{1}{n} \sum_{i=1}^n \sum_y [\mathbb{P}(Y = y \mid Z = Z_i) - \hat{\mathbb{P}}(Y = y \mid Z = Z_i)] \hat{a}_{yZ_i} \right|$$

$$\leq 2 \|\hat{a}\|_\infty \frac{1}{n} \sum_{i=1}^n \sum_y |\mathbb{P}(Y = y \mid Z = Z_i) - \hat{\mathbb{P}}(Y = y \mid Z = Z_i)| +$$

$$+ \|\tilde{a}_\varepsilon\|_\infty \frac{1}{n} \sum_{i=1}^n \sum_y |\mathbb{P}(Y = y \mid Z = Z_i) - \hat{\mathbb{P}}(Y = y \mid Z = Z_i)|$$

$$\leq 2 \|\hat{a}\|_\infty \sum_z \sum_y |\mathbb{P}(Y = y \mid Z = z) - \hat{\mathbb{P}}(Y = y \mid Z = z)| +$$

$$+ \|\tilde{a}_\varepsilon\|_\infty \sum_z \sum_y |\mathbb{P}(Y = y \mid Z = z) - \hat{\mathbb{P}}(Y = y \mid Z = z)|$$

$$\leq 4 \|\hat{a}\|_\infty \sum_z d_{\text{TV}}\left( \hat{P}_{Y|Z=z}, P_{Y|Z=z} \right) + 2 \|\tilde{a}_\varepsilon\|_\infty \sum_z d_{\text{TV}}\left( \hat{P}_{Y|Z=z}, P_{Y|Z=z} \right)$$

$$= \mathcal{O}_P(m^{-\lambda})$$

where the last equality is obtained using Assumptions 2.3 and 2.4, and the fact that $\|\tilde{a}_\varepsilon\|_\infty$ is tight (derived from $\sqrt{n}(\tilde{a}_\varepsilon - a^*_{u,\varepsilon}) = \mathcal{O}_P(1)$). Consequently,

$$\sqrt{n}(\tilde{U}_\varepsilon - \hat{U}_\varepsilon) = \sqrt{n}\mathcal{O}_P(m^{-\lambda}) = o(m^\lambda)\mathcal{O}_P(m^{-\lambda}) = o_P(1)$$

Finally, using Slutsky's theorem,

$$\sqrt{n}(\hat{U}_\varepsilon - U) = \sqrt{n}(\hat{U}_\varepsilon - \tilde{U}_\varepsilon) + \sqrt{n}(\tilde{U}_\varepsilon - U) \Rightarrow N(0, \text{Var} f_{u,\varepsilon}(X, Z, a^*_{u,\varepsilon}))$$

$\square$

*Proof of Corollary 3.1.* We prove the asymptotic distribution of $\sqrt{n}\left( \hat{p}_{u,\varepsilon} - p_{u,\varepsilon} \right)$. The result for the lower bound can be obtained analogously.

First, note that

$$\hat{\mathbb{P}}(h(X) = 1) - \mathbb{P}(h(X) = 1) = \mathcal{O}_P(m^{-1/2}) \tag{C.2}$$

by the standard central limit theorem.

Next, see that

$$\sqrt{n}\left(\hat{p}_{u,\varepsilon} - p_{u,\varepsilon}\right) =$$

$$= \sqrt{n}\left(\frac{\hat{U}_{\varepsilon}}{\hat{\mathbb{P}}(h(X)=1)} - \frac{U_{\varepsilon}}{\mathbb{P}(h(X)=1)}\right)$$

$$= \frac{1}{\hat{\mathbb{P}}(h(X)=1)}\sqrt{n}\left(\hat{U}_{\varepsilon} - \frac{\hat{\mathbb{P}}(h(X)=1)}{\mathbb{P}(h(X)=1)}U_{\varepsilon}\right)$$

$$= \frac{1}{\hat{\mathbb{P}}(h(X)=1)}\sqrt{n}\left(\hat{U}_{\varepsilon} - U_{\varepsilon}\right) + \frac{1}{\hat{\mathbb{P}}(h(X)=1)}\sqrt{n}\left(U_{\varepsilon} - \frac{\hat{\mathbb{P}}(h(X)=1)}{\mathbb{P}(h(X)=1)}U_{\varepsilon}\right)$$

$$= \frac{1}{\hat{\mathbb{P}}(h(X)=1)}\sqrt{n}\left(\hat{U}_{\varepsilon} - U_{\varepsilon}\right) + U_{\varepsilon}\sqrt{n}\left(\frac{1}{\hat{\mathbb{P}}(h(X)=1)} - \frac{1}{\mathbb{P}(h(X)=1)}\right)$$

$$= \frac{1}{\hat{\mathbb{P}}(h(X)=1)}\sqrt{n}\left(\hat{U}_{\varepsilon} - U_{\varepsilon}\right) + U_{\varepsilon}\sqrt{n}\left(\hat{\mathbb{P}}(h(X)=1) - \mathbb{P}(h(X)=1)\right)\left(\frac{-1}{\mathbb{P}(h(X)=1)^2}\right) + o_P\left(m^{-1/2}\right)$$

$$= \frac{1}{\hat{\mathbb{P}}(h(X)=1)}\sqrt{n}\left(\hat{U}_{\varepsilon} - U_{\varepsilon}\right) + o_P(1)$$

$$\Rightarrow N(0, \sigma_{p,u,\varepsilon}^2)$$

where the (i) fifth and sixth lines equality is obtained using Taylor's theorem, (ii) sixth and seventh lines equality is obtained using observation C.2 and the fact that $n = o(m^{(2\lambda)\wedge 1})$, and (iii) seventh to eighth lines equality is obtained using observation C.2, Theorem 2.5, and Slutsky's theorem.

We prove the asymptotic distribution of $\sqrt{n}(\hat{r}_{u,\varepsilon} - r_{u,\varepsilon})$. The result for the lower bound can be obtained analogously. From Lemma C.3, we know that there is an estimator $\hat{\mathbb{P}}(Y = 1)$ such that $\hat{\mathbb{P}}(Y = 1) - \mathbb{P}(Y = 1) = \mathcal{O}_P(m^{-(\lambda\wedge 1/2)})$, *i.e.*, it has enough precision. We use that estimator.

Next, see that

$$\sqrt{n}\left(\hat{r}_{u,\varepsilon} - r_{u,\varepsilon}\right) =$$

$$= \sqrt{n}\left(\frac{\hat{U}_{\varepsilon}}{\hat{\mathbb{P}}(Y=1)} - \frac{U_{\varepsilon}}{\mathbb{P}(Y=1)}\right)$$

$$= \frac{1}{\hat{\mathbb{P}}(Y=1)}\sqrt{n}\left(\hat{U}_{\varepsilon} - \frac{\hat{\mathbb{P}}(Y=1)}{\mathbb{P}(Y=1)}U_{\varepsilon}\right)$$

$$= \frac{1}{\hat{\mathbb{P}}(Y=1)}\sqrt{n}\left(\hat{U}_{\varepsilon} - U_{\varepsilon}\right) + \frac{1}{\hat{\mathbb{P}}(Y=1)}\sqrt{n}\left(U_{\varepsilon} - \frac{\hat{\mathbb{P}}(Y=1)}{\mathbb{P}(Y=1)}U_{\varepsilon}\right)$$

$$= \frac{1}{\hat{\mathbb{P}}(Y=1)}\sqrt{n}\left(\hat{U}_{\varepsilon} - U_{\varepsilon}\right) + U_{\varepsilon}\sqrt{n}\left(\frac{1}{\hat{\mathbb{P}}(Y=1)} - \frac{1}{\mathbb{P}(Y=1)}\right)$$

$$= \frac{1}{\hat{\mathbb{P}}(Y=1)}\sqrt{n}\left(\hat{U}_{\varepsilon} - U_{\varepsilon}\right) + U_{\varepsilon}\sqrt{n}\left(\hat{\mathbb{P}}(Y=1) - \mathbb{P}(Y=1)\right)\left(\frac{-1}{\mathbb{P}(Y=1)^2}\right) + o_P\left(m^{-1/2}\right)$$

$$= \frac{1}{\hat{\mathbb{P}}(Y=1)}\sqrt{n}\left(\hat{U}_{\varepsilon} - U_{\varepsilon}\right) + o_P(1)$$

$$\Rightarrow N(0, \sigma_{r,u,\varepsilon}^2)$$

Finally, we prove the asymptotic distribution of $\sqrt{n}(\hat{F}_{u,\varepsilon} - F_{u,\varepsilon})$. The result for the lower bound can be obtained analogously. From the facts stated above, we know that

$$\hat{\mathbb{P}}(h(X) = 1) + \hat{\mathbb{P}}(Y = 1) - [\mathbb{P}(h(X) = 1) + \mathbb{P}(Y = 1)] =$$

$$= [\hat{\mathbb{P}}(h(X) = 1) - \mathbb{P}(h(X) = 1)] + [\hat{\mathbb{P}}(Y = 1) - \mathbb{P}(Y = 1)] =$$

$$= \mathcal{O}_P(m^{-(\lambda\wedge 1/2)})$$

Then,

$$\sqrt{n}\left(\hat{F}_{u,\varepsilon} - F_{u,\varepsilon}\right) =$$

$$= \sqrt{n}\left(\frac{2\hat{U}_\varepsilon}{[\hat{\mathbb{P}}(h(X)=1)+\hat{\mathbb{P}}(Y=1)]} - \frac{2U_\varepsilon}{[\mathbb{P}(h(X)=1)+\mathbb{P}(Y=1)]}\right)$$

$$= \frac{2}{[\hat{\mathbb{P}}(h(X)=1)+\hat{\mathbb{P}}(Y=1)]}\sqrt{n}\left(\hat{U}_\varepsilon - \frac{[\hat{\mathbb{P}}(h(X)=1)+\hat{\mathbb{P}}(Y=1)]}{[\mathbb{P}(h(X)=1)+\mathbb{P}(Y=1)]}U_\varepsilon\right)$$

$$= \frac{2}{[\hat{\mathbb{P}}(h(X)=1)+\hat{\mathbb{P}}(Y=1)]}\sqrt{n}\left(\hat{U}_\varepsilon - U_\varepsilon\right) + \frac{1}{[\hat{\mathbb{P}}(h(X)=1)+\hat{\mathbb{P}}(Y=1)]}\sqrt{n}\left(2U_\varepsilon - \frac{[\hat{\mathbb{P}}(h(X)=1)+\hat{\mathbb{P}}(Y=1)]}{[\mathbb{P}(h(X)=1)+\mathbb{P}(Y=1)]}2U_\varepsilon\right)$$

$$= \frac{2}{[\hat{\mathbb{P}}(h(X)=1)+\hat{\mathbb{P}}(Y=1)]}\sqrt{n}\left(\hat{U}_\varepsilon - U_\varepsilon\right) + 2U_\varepsilon\sqrt{n}\left(\frac{1}{[\hat{\mathbb{P}}(h(X)=1)+\hat{\mathbb{P}}(Y=1)]} - \frac{1}{[\mathbb{P}(h(X)=1)+\mathbb{P}(Y=1)]}\right)$$

$$= \frac{2}{[\hat{\mathbb{P}}(h(X)=1)+\hat{\mathbb{P}}(Y=1)]}\sqrt{n}\left(\hat{U}_\varepsilon - U_\varepsilon\right) +$$

$$\quad + 2U_\varepsilon\sqrt{n}\left([\hat{\mathbb{P}}(h(X)=1)+\hat{\mathbb{P}}(Y=1)] - [\mathbb{P}(h(X)=1)+\mathbb{P}(Y=1)]\right)\left(\frac{-1}{[\mathbb{P}(h(X)=1)+\mathbb{P}(Y=1)]^2}\right) + o_P\left(m^{-1/2}\right)$$

$$= \frac{2}{[\hat{\mathbb{P}}(h(X)=1)+\hat{\mathbb{P}}(Y=1)]}\sqrt{n}\left(\hat{U}_\varepsilon - U_\varepsilon\right) + o_P(1)$$

$$\Rightarrow N(0, \sigma^2_{F,u,\varepsilon})$$

where all the steps are justified as before.

$\square$

## C.1 Auxiliary lemmas

**Lemma C.1.** *Define*

$$\tilde{f}_{u,\varepsilon}(x,z,a) \triangleq f_{u,\varepsilon}(x,z,a) + \sum_{z'\in\mathcal{Z}}\left(\sum_{y\in\mathcal{Y}} a_{yz'}\right)^2$$

*Then*

$$\inf_{a\in\mathbb{R}^{|\mathcal{Y}|\times|\mathcal{Z}|}} \mathbb{E}[\tilde{f}_{u,\varepsilon}(X,Z,a)] = \inf_{a\in\mathcal{A}} \mathbb{E}[f_{u,\varepsilon}(X,Z,a)]$$

*Proof.* First, see that

$$\mathbb{E}[\tilde{f}_{u,\varepsilon}(X,Z,a)] \geq \mathbb{E}[f_{u,\varepsilon}(X,Z,a)]$$

From Assumption 2.2, we know that there exists some $a^*_{u,\varepsilon} \in \mathbb{R}^{|\mathcal{Y}|\times|\mathcal{Z}|}$ such that

$$\inf_{a\in\mathcal{A}} \mathbb{E}[f_{u,\varepsilon}(X,Z,a)] = \mathbb{E}[f_{u,\varepsilon}(X,Z,a^*_{u,\varepsilon})]$$

For that specific $a^*_{u,\varepsilon}$, we have that

$$\mathbb{E}[\tilde{f}_{u,\varepsilon}(X,Z,a^*_{u,\varepsilon})] = \mathbb{E}[f_{u,\varepsilon}(X,Z,a^*_{u,\varepsilon})]$$

Consequently,

$$\inf_{a\in\mathbb{R}^{|\mathcal{Y}|\times|\mathcal{Z}|}} \mathbb{E}[\tilde{f}_{u,\varepsilon}(X,Z,a)] = \mathbb{E}[\tilde{f}_{u,\varepsilon}(X,Z,a^*_{u,\varepsilon})] = \mathbb{E}[f_{u,\varepsilon}(X,Z,a^*_{u,\varepsilon})] = \inf_{a\in\mathcal{A}} \mathbb{E}[f_{u,\varepsilon}(X,Z,a)]$$

$\square$

**Lemma C.2.** *The function of $a$ given by $\mathbb{E}[\tilde{f}_{u,\varepsilon}(X,Z,a)]$ has positive definite Hessian, i.e.,*

$$H_\varepsilon(a) = \nabla^2_a \mathbb{E}[\tilde{f}_{u,\varepsilon}(X,Z,a)] \succ 0$$

*and, consequently, it is strictly convex.*

*Proof.* See that

$$H_\varepsilon(a) = \nabla^2_a \mathbb{E}[f_{u,\varepsilon}(X,Z,a)] + \nabla^2_a\left[\sum_{z'\in\mathcal{Z}}\left(\sum_{y\in\mathcal{Y}} a_{yz'}\right)^2\right]$$

We start computing the first term in the sum.

First, for an arbitrary pair $(k,l) \in \mathcal{Y} \times \mathcal{Z}$, define

$$s_{kl}(x) \triangleq \frac{\exp\left(\frac{g(x,k,l)+a_{kl}}{\varepsilon}\right)}{\sum_y \exp\left(\frac{g(x,y,l)+a_{yl}}{\varepsilon}\right)}$$

Now, see that

$$\frac{\partial}{\partial a_{kl}} f_{u,\varepsilon}(x,z,a) = \mathbb{1}_{\{l\}}(z)\left[s_{kl}(x) - \mathbb{P}(Y=k \mid Z=l)\right]$$

and

$$\frac{\partial^2}{\partial a_{pl}\partial a_{kl}} f_{u,\varepsilon}(x,z,a) = \frac{1}{\varepsilon}\mathbb{1}_{\{l\}}(z)\left[\mathbb{1}_{\{k\}}(p)s_{kl}(x) - s_{kl}(x)s_{pl}(x)\right]$$

See that $\frac{\partial^2}{\partial a_{pb}\partial a_{kl}} f_{u,\varepsilon}(x,z,a) = 0$ if $b \neq l$. Consequently, the Hessian $\nabla_a^2 f_{u,\varepsilon}(x,z,a)$ is block diagonal.

Consequently, because the second derivatives are bounded, we can push them inside the expectations and get

$$\frac{\partial^2}{\partial a_{pl}\partial a_{kl}} \mathbb{E}\left[f_{u,\varepsilon}(X,Z,a)\right] =$$

$$= \mathbb{E}\left[\frac{\partial^2}{\partial a_{pl}\partial a_{kl}} f_{u,\varepsilon}(X,Z,a)\right]$$

$$= \frac{1}{\varepsilon}\mathbb{E}\left[\mathbb{1}_{\{l\}}(Z)\left[\mathbb{1}_{\{k\}}(p)s_{kl}(X) - s_{kl}(X)s_{pl}(X)\right]\right]$$

$$= \frac{1}{\varepsilon}\mathbb{1}_{\{k\}}(p) \cdot \mathbb{E}\left[\mathbb{P}(Z=l \mid X)s_{kl}(X)\right] - \frac{1}{\varepsilon}\mathbb{E}\left[\mathbb{P}(Z=l \mid X)s_{kl}(X)s_{pl}(X)\right]$$

Because $\nabla_a^2 f_{u,\varepsilon}(x,z,a)$ is block diagonal, we know that the Hessian

$$\nabla_a^2 \mathbb{E}\left[f_{u,\varepsilon}(X,Z,a)\right]$$

is block diagonal (one block for each segment $a_{\cdot z}$ of the vector $a$). Now, realize that

$$\nabla_a^2\left[\sum_{z'\in\mathcal{Z}}\left(\sum_{y\in\mathcal{Y}} a_{yz'}\right)^2\right]$$

is also block diagonal, with each block being matrices of ones. In this case, we also have one block for each segment $a_{\cdot z}$ of the vector $a$. Consequently, $H_\varepsilon(a)$ is block diagonal, and it is positive definite if and only if all of its blocks are positive definite. Let us analyse an arbitrary block of $H_\varepsilon(a)$, e.g., $\nabla_{a_{\cdot l}}^2 \mathbb{E}\left[\tilde{f}_{u,\varepsilon}(X,Z,a)\right]$. Let $\mathbf{s}_{\cdot l}(x)$ be the vector composed of $s_{kl}(x)$ for all $k$. If $\mathbb{1} \in \mathbb{R}^{|\mathcal{Y}|}$ denotes a vector of ones, then,

$$\nabla_{a_{\cdot l}}^2 \mathbb{E}\left[\tilde{f}_{u,\varepsilon}(X,Z,a)\right] =$$

$$= \frac{1}{\varepsilon}\mathrm{diag}\left(\mathbb{E}\left[\mathbb{P}(Z=l \mid X)\mathbf{s}_{\cdot l}(X)\right]\right) - \frac{1}{\varepsilon}\mathbb{E}\left[\mathbb{P}(Z=l \mid X)\mathbf{s}_{\cdot l}(X)\mathbf{s}_{\cdot l}(X)^\top\right] + \mathbb{1}\mathbb{1}^\top$$

$$= \mathbb{E}\left\{\frac{1}{\varepsilon}\mathbb{P}(Z=l \mid X)\left[\mathrm{diag}\left(\mathbf{s}_{\cdot l}(X)\right) - \mathbf{s}_{\cdot l}(X)\mathbf{s}_{\cdot l}(X)^\top + \frac{\varepsilon}{\mathbb{P}(Z=l)}\mathbb{1}\mathbb{1}^\top\right]\right\}$$

$$= \mathbb{E}\left\{\frac{1}{\varepsilon}\mathbb{P}(Z=l \mid X)\left[\mathrm{diag}\left(\mathbf{s}_{\cdot l}(X)\right) - \mathbf{s}_{\cdot l}(X)\mathbf{s}_{\cdot l}(X)^\top + \tilde{\mathbb{1}}\tilde{\mathbb{1}}^\top\right]\right\}$$

where $\tilde{\mathbb{1}} = \sqrt{\frac{\varepsilon}{\mathbb{P}(Z=l)}}\mathbb{1}$. See that $\mathrm{diag}\left(\mathbf{s}_{\cdot l}(x)\right)$ has rank $|\mathcal{Y}|$ (full rank) while $\mathbf{s}_{\cdot l}(x)\mathbf{s}_{\cdot l}(x)^\top$ is rank one for every $x \in \mathbb{R}^{d_X}$. Consequently, the rank of the difference

$$D(x) = \mathrm{diag}\left(\mathbf{s}_{\cdot l}(x)\right) - \mathbf{s}_{\cdot l}(x)\mathbf{s}_{\cdot l}(x)^\top$$

is greater or equal $|\mathcal{Y}| - 1$. It is the case that $\text{rank}(D(x)) = |\mathcal{Y}| - 1$ because $\tilde{\mathbb{1}}$ is in the null space of $D$:

$$D(x)\tilde{\mathbb{1}} = \left[\text{diag}\big(\mathbf{s}_{.l}(x)\big) - \mathbf{s}_{.l}(x)\mathbf{s}_{.l}(x)^\top\right]\tilde{\mathbb{1}} = \sqrt{\tfrac{\varepsilon}{\mathbb{P}(Z=l)}}(\mathbf{s}_{.l}(x) - \mathbf{s}_{.l}(x)) = 0$$

Moreover, the range of $D(x)$ and $\tilde{\mathbb{1}}\tilde{\mathbb{1}}^\top$ are orthogonal. For any two vectors $\mathbf{v}, \mathbf{u} \in \mathbb{R}^{|\mathcal{Y}|}$, we have that

$$(D(x)\mathbf{v})^\top(\tilde{\mathbb{1}}\tilde{\mathbb{1}}^\top\mathbf{u}) = \mathbf{v}^\top D(x)^\top\tilde{\mathbb{1}}\tilde{\mathbb{1}}^\top\mathbf{u} = \mathbf{v}^\top D(x)\tilde{\mathbb{1}}\tilde{\mathbb{1}}^\top\mathbf{u} = \mathbf{v}^\top 0\tilde{\mathbb{1}}^\top\mathbf{u} = 0$$

That implies $D(x) + \tilde{\mathbb{1}}\tilde{\mathbb{1}}^\top$ is full rank. To see that, let $\mathbf{v} \in \mathbb{R}^{|\mathcal{Y}|}$ be arbitrary and see

$$(D(x) + \tilde{\mathbb{1}}\tilde{\mathbb{1}}^\top)\mathbf{v} = 0 \Rightarrow D(x)\mathbf{v} = \tilde{\mathbb{1}}\tilde{\mathbb{1}}^\top\mathbf{v} = 0$$

Because $D(x)\mathbf{v} = 0$, it means that $\mathbf{v} = \theta\tilde{\mathbb{1}}$ for some constant $\theta$. If $\theta \neq 0$, then $\tilde{\mathbb{1}}\tilde{\mathbb{1}}^\top\mathbf{v} = \theta|\mathcal{Y}|\tilde{\mathbb{1}} \neq 0$. Therefore, $\theta = 0$ and $\mathbf{v} = 0$.

Now, let $\mathbf{u} \in \mathbb{R}^{|\mathcal{Y}|}$ be arbitrary non-null vector and see

$$\mathbf{u}^\top D(x)\mathbf{u} = \mathbf{u}^\top \text{diag}\big(\mathbf{s}_{.l}(x)\big)\mathbf{u} - \mathbf{u}^\top\mathbf{s}_{.l}(x)\mathbf{s}_{.l}(x)^\top\mathbf{u}$$

$$= \textstyle\sum_y u_y^2 s_{yl}(x) - \left(\sum_y u_y s_{yl}(x)\right)^2$$

$$= \left(\textstyle\sum_y s_{yl}(x)\right)\left(\sum_y u_y^2 s_{yl}(x)\right) - \left(\sum_y u_y s_{yl}(x)\right)^2$$

$$= \left(\textstyle\sum_y \sqrt{s_{yl}(x)}\sqrt{s_{yl}(x)}\right)\left(\sum_y \sqrt{u_y^2 s_{yl}(x)}\sqrt{u_y^2 s_{yl}(x)}\right) - \left(\sum_y u_y s_{yl}(x)\right)^2$$

$$\geq \left(\textstyle\sum_y u_y s_{yl}(x)\right)^2 - \left(\sum_y u_y s_{yl}(x)\right)^2 = 0$$

by the Cauchy–Schwarz inequality. Then, $D(x)$ is positive semidefinite, and because $\tilde{\mathbb{1}}\tilde{\mathbb{1}}^\top$ is also positive semidefinite, their sum needs to be positive definite for all $x$ (that matrix is full rank). Each block of $H_\varepsilon(a)$ is positive definite; consequently, $H_\varepsilon(a)$ is positive definite. $\qquad\square$

**Lemma C.3.** *Assume Assumption 2.4 holds. Let*

$$\hat{\mathbb{P}}(Y = 1) = \tfrac{1}{m}\textstyle\sum_{i=1}^m \mathbb{E}_{\hat{P}_{Y|Z}}[Y \mid Z = Z_i]$$

*Then*

$$\hat{\mathbb{P}}(Y = 1) - \mathbb{P}(Y = 1) = \mathcal{O}_P(m^{-(\lambda\wedge 1/2)})$$

*Proof.* To derive this result, see that

$$|\hat{\mathbb{P}}(Y = 1) - \mathbb{P}(Y = 1)| =$$

$$= \left|\tfrac{1}{m}\textstyle\sum_{i=1}^m \mathbb{E}_{P_{Y|Z}}[Y \mid Z = Z_i] - \mathbb{P}(Y = 1) + \tfrac{1}{m}\sum_{i=1}^m \mathbb{E}_{\hat{P}_{Y|Z}}[Y \mid Z = Z_i] - \mathbb{E}_{P_{Y|Z}}[Y \mid Z = Z_i]\right|$$

$$\leq \left|\tfrac{1}{m}\textstyle\sum_{i=1}^m \mathbb{E}_{P_{Y|Z}}[Y \mid Z = Z_i] - \mathbb{P}(Y = 1)\right| + \tfrac{1}{m}\sum_{i=1}^m \left|\mathbb{E}_{\hat{P}_{Y|Z}}[Y \mid Z = Z_i] - \mathbb{E}_{P_{Y|Z}}[Y \mid Z = Z_i]\right|$$

$$= \mathcal{O}_P(m^{-1/2}) + \tfrac{1}{m}\textstyle\sum_{i=1}^m \left|\mathbb{P}_{\hat{P}_{Y|Z}}[Y = 1 \mid Z = Z_i] - \mathbb{P}_{P_{Y|Z}}[Y = 1 \mid Z = Z_i]\right|$$

$$\leq \mathcal{O}_P(m^{-1/2}) + \textstyle\sum_{z\in\mathcal{Z}} \left|\mathbb{P}_{\hat{P}_{Y|Z}}[Y = 1 \mid Z = z] - \mathbb{P}_{P_{Y|Z}}[Y = 1 \mid Z = z]\right|$$

$$\leq \mathcal{O}_P(m^{-1/2}) + \textstyle\sum_{y\in\{0,1\}}\sum_{z\in\mathcal{Z}} \left|\mathbb{P}_{\hat{P}_{Y|Z}}[Y = y \mid Z = z] - \mathbb{P}_{P_{Y|Z}}[Y = y \mid Z = z]\right|$$

$$= \mathcal{O}_P(m^{-1/2}) + 2\textstyle\sum_{z\in\mathcal{Z}} d_{\text{TV}}\left(\hat{P}_{Y|Z=z}, P_{Y|Z=z}\right)$$

$$= \mathcal{O}_P(m^{-1/2}) + \mathcal{O}_P(m^{-\lambda})$$

$$= \mathcal{O}_P(m^{-(\lambda\wedge 1/2)})$$

where the standard central limit theorem obtains the third step, the sixth step is obtained by the formula of the total variation distance for discrete measures, and the seventh step is obtained by Assumption 2.4. $\qquad\square$

**Lemma C.4.** *For some $\kappa > 0$ and any $y \in \mathcal{Y}$ assume that $\kappa \leq p_y \leq 1 - \kappa$. Define $\mathcal{A} = \{a_y \in \mathbb{R} : \sum_y a_y = 0\}$. Then the optima of the following problems*

$$\inf_{a \in \mathcal{A}} f_u(a), \quad f_u(a) \triangleq \mathbb{E}\Big[\epsilon \log\Big\{\tfrac{1}{|\mathcal{Y}|} \sum_y \exp\big(\tfrac{g(X,y)+a_y}{\epsilon}\big)\Big\}\Big] - \sum_y p_y a_y\,,$$

$$\sup_{a \in \mathcal{A}} f_l(a), \quad f_l(a) \triangleq \mathbb{E}\Big[-\epsilon \log\Big\{\tfrac{1}{|\mathcal{Y}|} \sum_y \exp\big(\tfrac{g(X,y)+a_y}{-\epsilon}\big)\Big\}\Big] - \sum_y p_y a_y \tag{C.3}$$

*are attained in a compact set $K(\kappa, L) \subset \mathcal{A}$ where $\|g\|_\infty \leq L$.*

*Proof of lemma C.4.* We shall only prove this for the minimization problem. The conclusion for the maximization problem follows in a similar way.

**Strict convexity:** The second derivation of $f_u$ is

$$\nabla_{a_y, a_{y'}} f_u(a) = \tfrac{1}{\epsilon} \mathbb{E}\big[p(X, y, a)\{\delta_{y,y'} - p(X, y', a)\}\big] \tag{C.4}$$

where $p(x, y, a) \triangleq \dfrac{\exp\big(\frac{g(X,y)+a_y}{\epsilon}\big)}{\sum_{i \in \mathcal{Y}} \exp\big(\frac{g(X,i)+a_i}{\epsilon}\big)}$. For any $u \in \mathbb{R}^{\mathcal{Y}}$ we have

$$\begin{aligned}
u^\top \nabla^2 f_u(a) u &= \sum_{i,j \in \mathcal{Y}} u_i u_j \nabla_{a_i, a_j} f_u(a) \\
&= \tfrac{1}{\epsilon} \sum_{i,j \in \mathcal{Y}} u_i u_j \mathbb{E}\big[p(X, i, a)\{\delta_{i,j} - p(X, j, a)\}\big] \\
&= \tfrac{1}{\epsilon} \mathbb{E}\Big[\sum_i u_i^2 p(X, i, a) - \big\{\sum_i u_i p(X, i, a)\big\}^2\Big] = \tfrac{1}{\epsilon} \mathbb{E}[\sigma^2(X, u, a)] \geq 0
\end{aligned} \tag{C.5}$$

where $\sigma^2(x, u, a)$ is the variance of a categorical random variable taking the value $u_i$ with probability $p(x, i, a)$. This leads to the conclusion that the function is convex.

To establish strict convexity, we fix $a \in \mathcal{A}$ and notice that $0 < p(x, y, a) < 1$ (because $\|g\|_\infty \leq L$). Therefore, the $\sigma^2(x, u, a)$ can be zero only when the $u_i$'s are all equal. Since $\mathcal{A} = \{a \in \mathbb{R}^{\mathcal{Y}} : \sum_y a_y = 0\}$, such $u$ belongs to the space $\mathcal{A}$ in the unique case $u_i = 0$. This leads to the conclusion that for any $u \in \mathcal{A}$ and $u \neq 0$

$$u^\top \nabla^2 f_u(a) u = \tfrac{1}{\epsilon} \mathbb{E}[\sigma^2(X, u, a)] > 0\,.$$

Therefore, $f_u$ is strictly convex and has a unique minimizer in $\mathcal{A}$. In the remaining part of the proof, we focus on the first-order condition.

**First order condition:** The first-order condition is

$$h(a, y) = p_y \text{ for all } y \in \mathcal{Y}, \text{ where } h(a, y) \triangleq \mathbb{E}[p(X, y, a)] \tag{C.6}$$

To show that (C.6) has a solution in a compact ball $K(\epsilon, \kappa, L) \triangleq \{a \in \mathcal{A} : \|a\|_2 \leq M(\epsilon, \kappa, L)\}$ we construct an $M(\epsilon, \kappa, L) > 0$ such that $\min_y h(a, y) < \kappa$ whenever $\|a\|_2 > M(\epsilon, \kappa, L)$. Since $\kappa \leq p_y \leq 1 - \kappa$ for all $y \in \mathcal{Y}$ this concludes that the solution to (C.6) must be inside the $K(\epsilon, \kappa, L)$.

**Construction of the compact set:** Let us define $y_{\min} \triangleq \arg\min_y a_y$ and $y_{\max} \triangleq \arg\max_y a_y$. To construct $M(\kappa, L, \epsilon)$ (we shall write it simply as $M$ whenever it is convenient to do so) we notice that

$$\begin{aligned}
h(a, y_{\min}) &= \mathbb{E}[p(X, y_{\min}, a)] \\
&= \mathbb{E}\Big[\dfrac{\exp\big(\frac{g(X,1)+a_{y_{\min}}}{\epsilon}\big)}{\sum_{i \in \mathcal{Y}} \exp\big(\frac{g(X,i)+a_i}{\epsilon}\big)}\Big] \\
&\leq \dfrac{\exp\big(\frac{L+a_{y_{\min}}}{\epsilon}\big)}{\exp\big(\frac{L+a_{y_{\min}}}{\epsilon}\big) + \sum_{i \geq 2} \exp\big(\frac{-L+a_i}{\epsilon}\big)} \\
&= \dfrac{\exp\big(\frac{2L}{\epsilon}\big)}{\exp\big(\frac{2L}{\epsilon}\big) + \sum_{i \geq 2} \exp\big(\frac{a_i - a_{y_{\min}}}{\epsilon}\big)} \\
&\leq \dfrac{\exp\big(\frac{2L}{\epsilon}\big)}{\exp\big(\frac{2L}{\epsilon}\big) + \exp\big(\frac{a_{y_{\max}} - a_{y_{\min}}}{\epsilon}\big)} \leq \dfrac{\exp\big(\frac{2L}{\epsilon}\big)}{\exp\big(\frac{2L}{\epsilon}\big) + \exp\big(\frac{R}{\epsilon}\big)}\,,
\end{aligned} \tag{C.7}$$

where

$$R \triangleq \min\{\max_y a_y - \min_y a_y : \sum_i a_i = 0, \sum_i a_i^2 > M^2\}\,.$$

We rewrite the constraints of the optimization as:

$$R \triangleq \min \left\{ \max_y a_y - \min_y a_y : \text{mean}\{a_i\} = 0, \text{var}\{a_i\} > \frac{M^2}{|\mathcal{Y}|} \right\}.$$

where we use the Popoviciu's inequality on variance to obtain

$$\frac{M^2}{|\mathcal{Y}|} < \text{var}\{a_i\} \leq \frac{(\max_y a_y - \min_y a_y)^2}{4}, \quad \text{or} \quad \max_y a_y - \min_y a_y > \frac{2M}{\sqrt{|\mathcal{Y}|}},$$

Thus $R \geq \frac{2M}{\sqrt{|\mathcal{Y}|}}$ whenever $\|a\|_2 > M$. We use this inequality in (C.7) and obtain

$$h(a, y_{\min}) \leq \frac{\exp\left(\frac{2L}{\epsilon}\right)}{\exp\left(\frac{2L}{\epsilon}\right) + \exp\left(\frac{R}{\epsilon}\right)} \leq \frac{\exp\left(\frac{2L}{\epsilon}\right)}{\exp\left(\frac{2L}{\epsilon}\right) + \exp\left(\frac{2M}{\epsilon\sqrt{|\mathcal{Y}|}}\right)}.$$

Finally, we choose $M = M(\epsilon, \kappa, L) > 0$ large enough such that

$$h(a, y_{\min}) \leq \frac{\exp\left(\frac{2L}{\epsilon}\right)}{\exp\left(\frac{2L}{\epsilon}\right) + \exp\left(\frac{2M}{\epsilon\sqrt{|\mathcal{Y}|}}\right)} < \kappa.$$

For such an $M$ we concludes that $h(a, y_{\min}) < \kappa$ whenever $\|a\|_2 > M$.

$\square$

**Lemma C.5.** $L_\varepsilon$ and $U_\varepsilon$ are attained by some optimizers in a compact set $K(\epsilon, \kappa_z, z \in \mathcal{Z}; L) \supset \mathcal{A}$ ((2.2)), where $\kappa_z = \min\{p_{y|z}, 1 - p_{y|z} : y \in \mathcal{Y}\}$.

*Proof of lemma C.5.* We shall only prove the case of the minimization problem. The proof for the maximization problem uses a similar argument.

A decomposition of the minimization problem according to the values of $Z$ follows.

$$\min_{a \in \mathcal{A}} \mathbb{E}\left[\epsilon \log\left\{\frac{1}{|\mathcal{Y}|} \sum_{y \in \mathcal{Y}} \exp\left(\frac{g(X,y,Z) + a_{Y,Z}}{\epsilon}\right)\right\} - \mathbb{E}\left[a_{Y,Z} \mid Z\right]\right]$$

$$= \sum_z p_z \left\{ \min_{\substack{a_{\cdot,z} \in \mathbb{R}^{\mathcal{Y}} \\ \sum_y a_{y,z} = 0}} \mathbb{E}\left[\epsilon \log\left\{\frac{1}{|\mathcal{Y}|} \sum_{y \in \mathcal{Y}} \exp\left(\frac{g(X,y,z) + a_{Y,z}}{\epsilon}\right)\right\} \mid Z = z\right] - \mathbb{E}[a_{Y,z} \mid Z = z] \right\}$$

$$= \sum_z p_z \left\{ \min_{\substack{a_{\cdot,z} \in \mathbb{R}^{\mathcal{Y}} \\ \sum_y a_{y,z} = 0}} \mathbb{E}\left[\epsilon \log\left\{\frac{1}{|\mathcal{Y}|} \sum_{y \in \mathcal{Y}} \exp\left(\frac{g(X,y,z) + a_{Y,z}}{\epsilon}\right)\right\} \mid Z = z\right] - \sum_y a_{y,z} p_{y|z} \right\}$$

(C.8)

where $p_z = P(Z = z)$ and $p_{y|z} = P(Y = y \mid Z = z)$. We fix a $z$ and consider the corresponding optimization problem in the decomposition. Then according to lemma C.4 the optimal point is in a compact set $K(\epsilon, \kappa_z, L)$. Thus the optimal point of the full problem is in the Cartesian product $\prod_z K(\epsilon, \kappa_z, L) \subset \mathcal{A}$, which a compact set. Thus we let $K(\epsilon, \kappa_z, z \in \mathcal{Z}; L) = \prod_z K(\epsilon, \kappa_z, L)$.

$\square$

**Lemma C.6.** *The optimizers for the problems in (2.3) are tight with respect to $m, n \to \infty$.*

*Proof of the lemma C.6.* Let $p_{y|z} = P(Y = y \mid Z = z)$ and $\hat{p}_{y|z}^{(m)} = \hat{P}_{Y|Z=z}^{(m)}(y)$. Since

$$\tfrac{1}{2} \sum_y |\hat{p}_{y|z}^{(m)} - p_{y|z}| = d_{\text{TV}}\left(\hat{P}_{Y|Z=z}^{(m)}, P_{Y|Z=z}\right) = \mathcal{O}_P(m^{-\lambda})$$

according to the assumption 2.4, for sufficiently large $m$ with high probability $p(m)$ ($p(m) \to 1$ as $m \to \infty$) $\frac{\kappa_z}{2} \leq \hat{p}_{y|z}^{(m)} \leq 1 - \frac{\kappa_z}{2}$ for all $y$. Fix such an $m$ and $\hat{p}_{y|z}^{(m)}$. In lemma C.5 we replace $P_{X,Z}$ with $\sum_{i=1}^n \delta_{X_i,Z_i}$ and $p_{y|z}$ with $\hat{p}_{y|z}^{(m)}$ to reach the conclusion that the optimizer is in the compact set $K(\epsilon, \kappa_z/2, z \in \mathcal{Z}; L)$. Thus, for sufficiently large $m$ and any $n$

$$\mathbb{P}\big(\text{optimizer is in } K(\epsilon, \kappa_z/2, z \in \mathcal{Z}; L)\big) \geq p(m).$$

This establishes that the optimizers are tight.

$\square$

# D  Proof of extra results

*Proof of Theorem 5.1.* We only prove the theorem for upper Fréchet bound. The proof of lower bound is similar.

First, notice that

$$
\begin{aligned}
\check{f}_{u,\varepsilon}(x,z,a) - f_{u,\epsilon}(x,z,a) &= \\
&= \varepsilon \log \left[ \tfrac{1}{|\mathcal{Y}|} \sum_{y\in\mathcal{Y}} \exp \left( \tfrac{g(x,y,z)+a_{yz}}{\varepsilon} \right) \right] - \mathbb{E}_{Q_{Y|Z}}[a_{Yz} \mid Z=z] \\
&\quad - \varepsilon \log \left[ \tfrac{1}{|\mathcal{Y}|} \sum_{y\in\mathcal{Y}} \exp \left( \tfrac{g(x,y,z)+a_{yz}}{\varepsilon} \right) \right] + \mathbb{E}_{P_{Y|Z}}[a_{Yz} \mid Z=z] \\
&= \mathbb{E}_{P_{Y|Z}}[a_{Yz} \mid Z=z] - \mathbb{E}_{Q_{Y|Z}}[a_{Yz} \mid Z=z]
\end{aligned}
$$

and thus

$$
\begin{aligned}
&|\check{f}_{u,\varepsilon}(x,z,a) - f_{u,\epsilon}(x,z,a)| && \text{(D.1)} \\
&= \left| \mathbb{E}_{P_{Y|Z}}[a_{Yz} \mid Z=z] - \mathbb{E}_{Q_{Y|Z}}[a_{Yz} \mid Z=z] \right| && \text{(D.2)} \\
&\le \|a\|_\infty \times 2 d_{\mathrm{TV}}\left( Q_{Y|Z=z}, P_{Y|Z=z} \right) && \text{(using } a^\top b \le \|a\|_\infty \|b\|_1 \text{)} && \text{(D.3)} \\
&\le 2\|a\|_\infty \delta \, . && \text{(by assumption)} && \text{(D.4)}
\end{aligned}
$$

Defining

$$
a^\star \triangleq \arg\min_a \mathbb{E}[f_{u,\varepsilon}(X,Z,a)], \quad \check{a} \triangleq \arg\min_a \mathbb{E}[\check{f}_{u,\varepsilon}(X,Z,a)] \,, \tag{D.5}
$$

we now establish that

$$
|\check{U}_\epsilon - U_\epsilon| \le 2\delta \max(\|a^\star\|_\infty, \|\check{a}\|_\infty) \,. \tag{D.6}
$$

This is easily established from the following arguments:

$$
\begin{aligned}
\check{U}_\epsilon &= \mathbb{E}[\check{f}_{u,\varepsilon}(X,Z,\check{a})] \\
&\le \mathbb{E}[\check{f}_{u,\varepsilon}(X,Z,a^\star)] && (\check{a} \text{ is the minimizer}) \\
&\le \mathbb{E}[f_{u,\varepsilon}(X,Z,a^\star)] + 2\|a^\star\|_\infty \delta && \text{(using eq. (D.4))} \\
&= U_\epsilon + 2\|a^\star\|_\infty \delta
\end{aligned} \tag{D.7}
$$

and similarly

$$
U_\epsilon = \mathbb{E}[f_{u,\varepsilon}(X,Z,a^\star)] \le \mathbb{E}[f_{u,\varepsilon}(X,Z,\check{a})] \le \mathbb{E}[\check{f}_{u,\varepsilon}(X,Z,\check{a})] + 2\|\check{a}\|_\infty \delta = \check{U}_\epsilon + 2\|a^\star\|_\infty \delta \,. \tag{D.8}
$$

By assumption, $a^\star$ and $\check{a}$ are bounded. Thus, we can find a $C > 0$, which is independent of $\delta > 0$, such that

$$
2\max(\|a^\star\|_\infty, \|\check{a}\|_\infty) \le C
$$

and the theorem holds for this $C$. $\qquad\square$

*Proof of Theorem 5.2.* For clarity, we shall assume $X$ is a continuous random variable in this proof, even though this is not strictly needed.

Let $Q_{X,Y,Z}^{(1)}$ and $Q_{X,Y,Z}^{(2)}$ be two distributions in $\Pi$, *i.e.*, with marginals $P_{X,Z}$ and $P_{Y,Z}$. Let the densities of $Q_{X,Y,Z}^{(1)}$ and $Q_{X,Y,Z}^{(2)}$ as $q_{X,Y,Z}^{(1)}$ and $q_{X,Y,Z}^{(2)}$. Then, see that

$$\left| \int g \mathrm{d}Q_{X,Y,Z}^{(1)} - \int g \mathrm{d}Q_{X,Y,Z}^{(2)} \right| =$$

$$= \left| \int \sum_{y,z} g(x,y,z)(q_{X,Y,Z}^{(1)}(x,y,z) - q_{X,Y,Z}^{(2)}(x,y,z)) \mathrm{d}x \right|$$

$$\leq \int \sum_{y,z} |g(x,y,z)| \cdot |q_{X,Y,Z}^{(1)}(x,y,z) - q_{X,Y,Z}^{(2)}(x,y,z)| \mathrm{d}x$$

$$\leq 2 \|g\|_\infty \sum_z p_Z(z) \tfrac{1}{2} \int \sum_y \cdot |q_{X,Y|Z}^{(1)}(x,y \mid z) - q_{X,Y|Z}^{(2)}(x,y \mid z)| \mathrm{d}x$$

$$= 2 \|g\|_\infty \mathbb{E} \left[ d_{\mathrm{TV}}(Q_{X,Y|Z}^{(1)}, Q_{X,Y|Z}^{(2)}) \right]$$

$$\leq 2 \|g\|_\infty \left\{ \mathbb{E} \left[ d_{\mathrm{TV}}(Q_{X,Y|Z}^{(1)}, P_{X|Z}P_{Y|Z}) \right] + \mathbb{E} \left[ d_{\mathrm{TV}}(Q_{X,Y|Z}^{(2)}, P_{X|Z}P_{Y|Z}) \right] \right\} \tag{D.9}$$

$$\leq 2 \|g\|_\infty \left\{ \mathbb{E} \left[ \sqrt{\tfrac{1}{2}\mathrm{KL}(Q_{X,Y|Z}^{(1)}||P_{X|Z}P_{Y|Z})} \right] + \mathbb{E} \left[ \sqrt{\tfrac{1}{2}\mathrm{KL}(Q_{X,Y|Z}^{(2)}||P_{X|Z}P_{Y|Z})} \right] \right\} \tag{D.10}$$

$$\leq 2 \|g\|_\infty \left\{ \sqrt{\tfrac{1}{2}\mathbb{E} \left[ \mathrm{KL}(Q_{X,Y|Z}^{(1)}||P_{X|Z}P_{Y|Z}) \right]} + \sqrt{\tfrac{1}{2}\mathbb{E} \left[ \mathrm{KL}(Q_{X,Y|Z}^{(2)}||P_{X|Z}P_{Y|Z}) \right]} \right\} \tag{D.11}$$

$$\leq 2 \|g\|_\infty \left\{ \sqrt{\tfrac{1}{2}H(X \mid Z)} + \sqrt{\tfrac{1}{2}H(X \mid Z)} \right\} \tag{D.12}$$

$$\leq \sqrt{8 \|g\|_\infty^2 H(X \mid Z)}$$

where step D.9 is justified by triangle inequality, step D.10 is justified by Pinsker's inequality, step D.11 is justified by Jensen's inequality, and step D.12 is justified by the fact that both expected KL terms are conditional mutual information terms, which can be bounded by the conditional entropy. Following the same idea, we can show that $\left| \int g \mathrm{d}Q_{X,Y,Z}^{(1)} - \int g \mathrm{d}Q_{X,Y,Z}^{(2)} \right| \leq \sqrt{8 \|g\|_\infty^2 H(Y \mid Z)}$. Consequently,

$$\left| \int g \mathrm{d}Q_{X,Y,Z}^{(1)} - \int g \mathrm{d}Q_{X,Y,Z}^{(2)} \right| \leq \min \left( \sqrt{8 \|g\|_\infty^2 H(X \mid Z)}, \sqrt{8 \|g\|_\infty^2 H(Y \mid Z)} \right)$$

$$= \sqrt{8 \|g\|_\infty^2 \min \left( H(X \mid Z), H(Y \mid Z) \right)}.$$

Because this statement is valid for any distributions $Q_{X,Y,Z}^{(1)}$ and $Q_{X,Y,Z}^{(2)}$ in $\Pi$, we have that

$$U - L \leq \sqrt{8 \|g\|_\infty^2 \min \left( H(X \mid Z), H(Y \mid Z) \right)}$$

$\square$

# E  Model selection using performance bounds

In this section, we propose and empirically evaluate three strategies for model selection using our estimated bounds in Equation 2.3.

### E.0.1  Introducing model selection strategies

Assume, for example, $g(x, y, z) = \mathbb{1}[h(x) = y]$ for a given classifier $h$, *i.e.*, we conduct model selection based on accuracy, even though we can easily extend the same idea to different choices of metrics, such as F1 score. The model selection problem consists of choosing the best model from a set $\mathcal{H} \triangleq \{h_1, \cdots, h_K\}$ in order to maximize out-of-sample accuracy. Define $\hat{L}_\varepsilon(h)$ and $\hat{U}_\varepsilon(h)$ as the estimated accuracy lower and upper bounds for a certain model $h$. The first strategy is to choose the model with *highest* accuracy lower bound, *i.e.*,

$$h^*_{\text{lower}} = \arg\max_{h_k \in \mathcal{H}} \hat{L}_\varepsilon(h_k)$$

Maximizing the accuracy lower bound approximates the distributionally robust optimization (DRO) [9] solution when the uncertainty set is given by $\Pi$ in 1.1. That is, we optimize for the worst-case distribution in the uncertainty set. Analogously, we can choose the model that optimizes the best-case scenario

$$h^*_{\text{upper}} = \arg\max_{h_k \in \mathcal{H}} \hat{U}_\varepsilon(h_k).$$

Moreover, if we want to guarantee that both worst and best-case scenarios are not bad, we can optimize the average of upper and lower bounds, *i.e.*,

$$h^*_{\text{avg}} = \arg\max_{h_k \in \mathcal{H}} \frac{\hat{L}_\varepsilon(h_k) + \hat{U}_\varepsilon(h_k)}{2}.$$

### E.0.2  Experiment setup

In this experiment, we select multilayer-perceptrons (MLPs). The considered MLPs have one hidden layer with a possible number of neurons in $\{50, 100\}$. Training is carried out with Adam [26], with possible learning rates in $\{.1, .001\}$ and weight decay ($l_2$ regularization parameter) in $\{.1, .001\}$. For those datasets that use the F1 score as the evaluation metric, we also tune the classification threshold in $\{.2, .4, .5, .6, .8\}$ (otherwise, they return the most probable class as a prediction). In total, $\mathcal{H}$ is composed of 8 trained models when evaluating accuracy and 40 models when evaluating the F1 score. We also consider directly using the label model (Snorkel [47]) to select models. For example, when the metric considered is accuracy, *i.e.*, we use

$$h^*_{\text{label\_model}} = \arg\max_{h_k \in \mathcal{H}} \frac{1}{n} \sum_{i=1}^n \mathbb{E}_{\hat{P}_{Y|Z}} \mathbb{1}[h_k(X) = Y \mid Z = Z_i],$$

which is a natural choice when $X \perp\!\!\!\perp Y \mid Z$. As benchmarks, we consider having a few labeled samples.

In Table 3, we report the average test scores of the chosen models over 10 repetitions for different random seeds (standard deviation report as subscript). The main message here is that, for those datasets in which our uncertainty about the score (given by the different upper and lower bounds) is small, *e.g.*, "'commercial" and "tennis", using our approaches leads to much better results when compared to using small sample sizes.

Now, we explore a different way of comparing models. Instead of making an explicit model selection, this experiment uses the proposed metrics, *i.e.*, accuracy lower/upper bounds, bounds average, to rank MLP classifiers. We rank the models using both the test set accuracy/F1 score (depending on Zhang et al. [63]) and alternative metrics, *i.e.*, accuracy/F1 score lower/upper bounds, bounds average, label model, and small labeled sample sizes. Then, we calculate a Pearson correlation between rankings and display numbers in Table 4. If the numbers are higher, it means that the proposed selection method is capable of distinguishing good from bad models. Table 4 shows that the bounds average and label model methods usually return the best results when no labels are used. Moreover, in some cases using a small labeled sample for model selection can relatively hurt performance (when the validation set is small, there is a chance all models will have the same or similar performances, leading to smaller or null rank correlations).

Table 3: Performance of selected models

| | metric | Lower bound | Upper bound | Bounds avg | Label model | Labeled ($n=10$) | Labeled ($n=25$) | Labeled ($n=50$) | Labeled ($n=100$) |
|---|---|---|---|---|---|---|---|---|---|
| agnews | acc | $0.77_{\pm0.00}$ | $0.78_{\pm0.00}$ | $0.78_{\pm0.00}$ | $0.77_{\pm0.00}$ | $0.75_{\pm0.02}$ | $0.75_{\pm0.02}$ | $0.75_{\pm0.03}$ | $0.77_{\pm0.00}$ |
| trec | acc | $0.27_{\pm0.01}$ | $0.29_{\pm0.02}$ | $0.29_{\pm0.02}$ | $0.29_{\pm0.02}$ | $0.28_{\pm0.02}$ | $0.28_{\pm0.02}$ | $0.28_{\pm0.02}$ | $0.28_{\pm0.02}$ |
| semeval | acc | $0.32_{\pm0.01}$ | $0.26_{\pm0.01}$ | $0.32_{\pm0.01}$ | $0.32_{\pm0.01}$ | $0.30_{\pm0.02}$ | $0.29_{\pm0.03}$ | $0.31_{\pm0.02}$ | $0.31_{\pm0.02}$ |
| chemprot | acc | $0.40_{\pm0.00}$ | $0.38_{\pm0.00}$ | $0.40_{\pm0.00}$ | $0.40_{\pm0.00}$ | $0.40_{\pm0.01}$ | $0.40_{\pm0.01}$ | $0.40_{\pm0.00}$ | $0.39_{\pm0.01}$ |
| youtube | acc | $0.89_{\pm0.02}$ | $0.84_{\pm0.05}$ | $0.87_{\pm0.01}$ | $0.89_{\pm0.02}$ | $0.90_{\pm0.02}$ | $0.90_{\pm0.02}$ | $0.90_{\pm0.02}$ | $0.91_{\pm0.02}$ |
| imdb | acc | $0.72_{\pm0.00}$ | $0.74_{\pm0.00}$ | $0.73_{\pm0.00}$ | $0.73_{\pm0.00}$ | $0.69_{\pm0.05}$ | $0.71_{\pm0.02}$ | $0.73_{\pm0.01}$ | $0.72_{\pm0.01}$ |
| yelp | acc | $0.81_{\pm0.00}$ | $0.81_{\pm0.00}$ | $0.81_{\pm0.00}$ | $0.81_{\pm0.00}$ | $0.81_{\pm0.04}$ | $0.81_{\pm0.03}$ | $0.81_{\pm0.01}$ | $0.82_{\pm0.01}$ |
| census | F1 | $0.49_{\pm0.00}$ | $0.18_{\pm0.00}$ | $0.49_{\pm0.00}$ | $0.49_{\pm0.00}$ | $0.49_{\pm0.00}$ | $0.49_{\pm0.00}$ | $0.49_{\pm0.00}$ | $0.50_{\pm0.00}$ |
| tennis | F1 | $0.76_{\pm0.01}$ | $0.76_{\pm0.01}$ | $0.76_{\pm0.01}$ | $0.75_{\pm0.01}$ | $0.71_{\pm0.01}$ | $0.71_{\pm0.01}$ | $0.71_{\pm0.01}$ | $0.71_{\pm0.02}$ |
| sms | F1 | $0.00_{\pm0.00}$ | $0.14_{\pm0.02}$ | $0.14_{\pm0.02}$ | $0.14_{\pm0.02}$ | $0.00_{\pm0.00}$ | $0.00_{\pm0.00}$ | $0.00_{\pm0.00}$ | $0.00_{\pm0.00}$ |
| cdr | F1 | $0.48_{\pm0.00}$ | $0.29_{\pm0.00}$ | $0.48_{\pm0.00}$ | $0.48_{\pm0.00}$ | $0.47_{\pm0.00}$ | $0.47_{\pm0.00}$ | $0.47_{\pm0.00}$ | $0.47_{\pm0.00}$ |
| basketball | F1 | $0.16_{\pm0.01}$ | $0.27_{\pm0.01}$ | $0.26_{\pm0.01}$ | $0.17_{\pm0.01}$ | $0.20_{\pm0.05}$ | $0.17_{\pm0.03}$ | $0.16_{\pm0.01}$ | $0.16_{\pm0.01}$ |
| spouse | F1 | $0.27_{\pm0.01}$ | $0.27_{\pm0.01}$ | $0.27_{\pm0.01}$ | $0.29_{\pm0.01}$ | $0.00_{\pm0.00}$ | $0.00_{\pm0.00}$ | $0.00_{\pm0.00}$ | $0.10_{\pm0.12}$ |
| commercial | F1 | $0.96_{\pm0.00}$ | $0.96_{\pm0.00}$ | $0.96_{\pm0.00}$ | $0.96_{\pm0.00}$ | $0.90_{\pm0.01}$ | $0.90_{\pm0.02}$ | $0.91_{\pm0.01}$ | $0.91_{\pm0.01}$ |

Table 4: Ranking correlation when ranking using test set and alternative metric

| | metric | Lower bound | Upper bound | Bounds avg | Label model | Labeled ($n=10$) | Labeled ($n=25$) | Labeled ($n=50$) | Labeled ($n=100$) |
|---|---|---|---|---|---|---|---|---|---|
| agnews | acc | $0.91_{\pm0.01}$ | $0.86_{\pm0.00}$ | $0.86_{\pm0.00}$ | $0.92_{\pm0.01}$ | $0.39_{\pm0.31}$ | $0.35_{\pm0.54}$ | $0.50_{\pm0.53}$ | $0.76_{\pm0.19}$ |
| trec | acc | $-0.32_{\pm0.36}$ | $0.37_{\pm0.32}$ | $0.19_{\pm0.34}$ | $0.44_{\pm0.39}$ | $-0.03_{\pm0.55}$ | $-0.06_{\pm0.39}$ | $-0.10_{\pm0.34}$ | $0.17_{\pm0.52}$ |
| semeval | acc | $0.87_{\pm0.04}$ | $-0.84_{\pm0.04}$ | $0.88_{\pm0.04}$ | $0.88_{\pm0.04}$ | $0.13_{\pm0.64}$ | $0.17_{\pm0.69}$ | $0.64_{\pm0.32}$ | $0.62_{\pm0.41}$ |
| chemprot | acc | $0.75_{\pm0.08}$ | $-0.66_{\pm0.11}$ | $0.38_{\pm0.11}$ | $0.49_{\pm0.15}$ | $-0.01_{\pm0.47}$ | $-0.12_{\pm0.45}$ | $-0.02_{\pm0.43}$ | $0.06_{\pm0.41}$ |
| youtube | acc | $0.38_{\pm0.21}$ | $-0.42_{\pm0.19}$ | $-0.01_{\pm0.21}$ | $0.17_{\pm0.29}$ | $0.41_{\pm0.35}$ | $0.42_{\pm0.27}$ | $0.38_{\pm0.31}$ | $0.58_{\pm0.19}$ |
| imdb | acc | $0.84_{\pm0.01}$ | $0.76_{\pm0.00}$ | $0.71_{\pm0.02}$ | $0.92_{\pm0.01}$ | $0.17_{\pm0.49}$ | $0.40_{\pm0.46}$ | $0.61_{\pm0.28}$ | $0.65_{\pm0.30}$ |
| yelp | acc | $0.01_{\pm0.08}$ | $-0.50_{\pm0.05}$ | $-0.49_{\pm0.05}$ | $-0.31_{\pm0.10}$ | $0.08_{\pm0.65}$ | $0.09_{\pm0.53}$ | $0.14_{\pm0.45}$ | $0.45_{\pm0.30}$ |
| census | F1 | $0.85_{\pm0.00}$ | $0.94_{\pm0.00}$ | $0.96_{\pm0.00}$ | $0.98_{\pm0.00}$ | $0.00_{\pm0.00}$ | $0.00_{\pm0.00}$ | $0.03_{\pm0.09}$ | $0.18_{\pm0.17}$ |
| tennis | F1 | $0.83_{\pm0.02}$ | $0.74_{\pm0.03}$ | $0.80_{\pm0.02}$ | $0.90_{\pm0.03}$ | $0.33_{\pm0.24}$ | $0.41_{\pm0.22}$ | $0.51_{\pm0.12}$ | $0.52_{\pm0.12}$ |
| sms | F1 | $0.00_{\pm0.00}$ | $0.99_{\pm0.02}$ | $0.99_{\pm0.02}$ | $1.00_{\pm0.00}$ | $0.00_{\pm0.00}$ | $0.00_{\pm0.00}$ | $0.00_{\pm0.00}$ | $0.00_{\pm0.00}$ |
| cdr | F1 | $0.80_{\pm0.00}$ | $0.89_{\pm0.00}$ | $0.92_{\pm0.00}$ | $0.98_{\pm0.00}$ | $0.03_{\pm0.08}$ | $0.07_{\pm0.11}$ | $0.07_{\pm0.11}$ | $0.09_{\pm0.11}$ |
| basketball | F1 | $0.85_{\pm0.01}$ | $0.69_{\pm0.00}$ | $0.73_{\pm0.01}$ | $0.81_{\pm0.00}$ | $0.35_{\pm0.29}$ | $0.53_{\pm0.18}$ | $0.59_{\pm0.00}$ | $0.59_{\pm0.00}$ |
| spouse | F1 | $0.53_{\pm0.00}$ | $0.86_{\pm0.00}$ | $0.86_{\pm0.00}$ | $1.00_{\pm0.00}$ | $0.00_{\pm0.00}$ | $0.00_{\pm0.00}$ | $0.00_{\pm0.00}$ | $0.10_{\pm0.12}$ |
| commercial | F1 | $0.95_{\pm0.00}$ | $0.96_{\pm0.00}$ | $0.99_{\pm0.00}$ | $1.00_{\pm0.00}$ | $0.14_{\pm0.13}$ | $0.26_{\pm0.14}$ | $0.28_{\pm0.13}$ | $0.31_{\pm0.06}$ |

# F  More on experiments

## F.1  Extra results for the Wrench experiment

Extra results for binary classification datasets can be found in Figures 4 and 5. In Table 5, we can see the results for multinomial classification datasets.

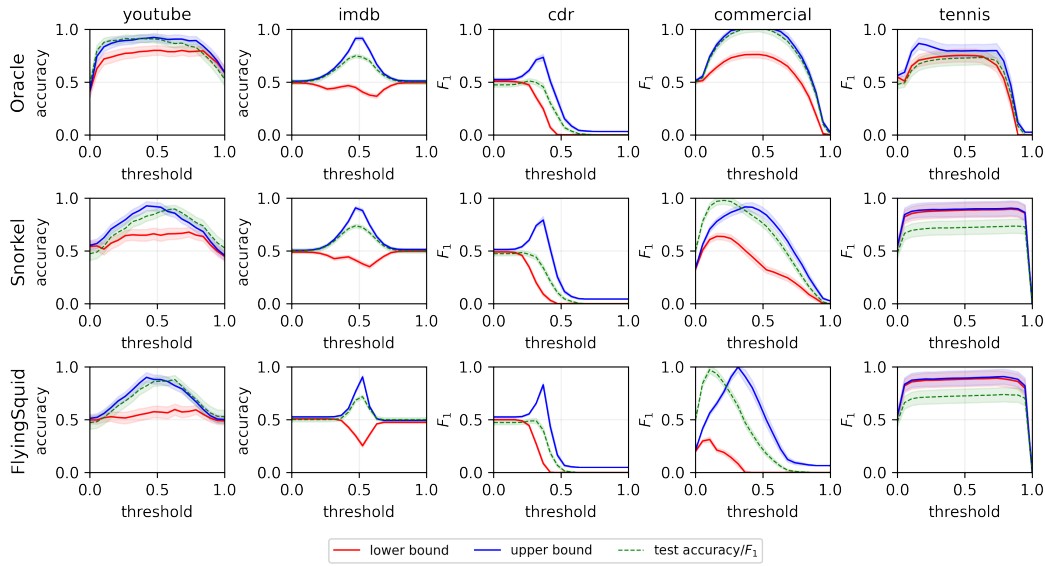

Figure 4: Bounds on classifier accuracies across classification thresholds for the Wrench datasets. Despite potential misspecification in Snorkel's and FlyingSquid's label model, it performs comparably to using labels to estimate $P_{Y|Z}$, giving approximate but meaningful bounds. .

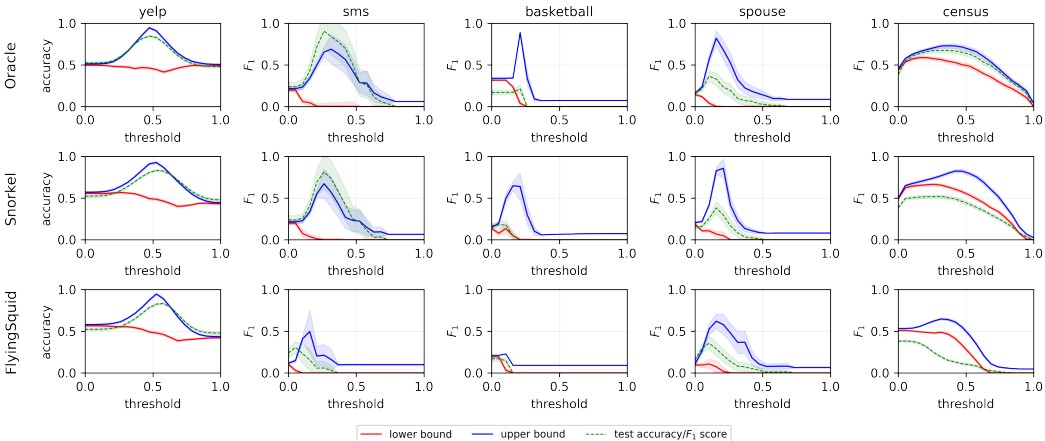

Figure 5: Bounds on classifier accuracies and F1 scores across classification thresholds for the Wrench datasets (using the full set of weak labels). Despite potential misspecification in Snorkel's and FlyingSquid's label model, it performs comparably to using labels to estimate $P_{Y|Z}$, giving approximate but meaningful bounds.

### F.2 Extra plots for the hate speech detection experiment

In Table 6, we can see the same results already in the main text plus the results for FlyingSquid.

## G Computing resources

All experiments were conducted using a virtual machine with 32 cores. The experiments are not computationally intensive and everything can be run within a few hours.

Table 5: Bounding the accuracy of classifiers in multinomial classification

| Dataset | Label model | Lower bound | Upper bound | Test accuracy |
|---|---|---|---|---|
| agnews | Oracle | $0.46_{\pm0.01}$ | $0.95_{\pm0.01}$ | $0.80_{\pm0.01}$ |
| agnews | Snorkel | $0.42_{\pm0.01}$ | $0.9_{\pm0.01}$ | $0.76_{\pm0.01}$ |
| agnews | FlyingSquid | $0.12_{\pm0.0}$ | $0.61_{\pm0.01}$ | $0.76_{\pm0.01}$ |
| trec | Oracle | $0.34_{\pm0.04}$ | $0.83_{\pm0.03}$ | $0.68_{\pm0.04}$ |
| trec | Snorkel | $0.31_{\pm0.04}$ | $0.70_{\pm0.03}$ | $0.47_{\pm0.04}$ |
| trec | FlyingSquid | $0.07_{\pm0.02}$ | $0.29_{\pm0.02}$ | $0.27_{\pm0.04}$ |
| semeval | Oracle | $0.54_{\pm0.04}$ | $0.78_{\pm0.03}$ | $0.72_{\pm0.04}$ |
| semeval | Snorkel | $0.36_{\pm0.03}$ | $0.70_{\pm0.03}$ | $0.56_{\pm0.04}$ |
| semeval | FlyingSquid | $0.12_{\pm0.0}$ | $0.14_{\pm0.0}$ | $0.32_{\pm0.04}$ |
| chemprot | Oracle | $0.43_{\pm0.02}$ | $0.75_{\pm0.02}$ | $0.60_{\pm0.02}$ |
| chemprot | Snorkel | $0.37_{\pm0.02}$ | $0.73_{\pm0.02}$ | $0.49_{\pm0.02}$ |
| chemprot | FlyingSquid | $0.05_{\pm0.0}$ | $0.23_{\pm0.01}$ | $0.46_{\pm0.02}$ |

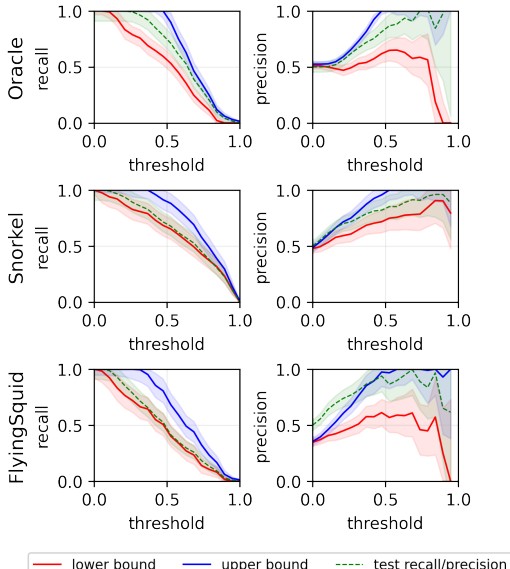

Figure 6: Precision and recall bounds for hate speech detection. These plots guide practitioners to trade off recall and precision in the absence of high-quality labels.

### G.1 Examples of prompts used in Section 4.2

**Prompt 1**

```
You should classify the target sentence as "spam" or "ham".
If definitions or examples are introduced, you should consider
them when classifying sentences. Respond with "spam" or "ham".

Target sentence: if your like drones, plz subscribe to Kamal Tayara.
He takes videos with  his drone that are absolutely beautiful. -- Response:
```

**Prompt 2**

```
You should classify the target sentence as "spam" or "ham".
If definitions or examples are introduced, you should consider
them when classifying sentences. Respond with "spam" or "ham".
```

Definition of spam: spam is a term referencing a broad category of
postings which abuse web−based
forms to post unsolicited advertisements as comments on forums,
blogs, wikis and online guestbook.

Definition of ham: texts that are not spam.

Target sentence: if your like drones, plz subscribe to Kamal Tayara.
He takes videos with  his drone that are absolutely beautiful. −− Response:

**Prompt 3**

You should classify the target sentence as "spam" or "ham".
If definitions or examples are introduced,
you should consider
them when classifying sentences.
Respond with "spam" or "ham".

Example 0: 860,000,000 lets make it first female to reach
one billion!! Share it and replay it!  −− Response: ham

Example 1: Waka waka eh eh −− Response: ham

Example 2: You guys should check out this EXTRAORDINARY website called
ZONEPA.COM . You can make money online and start working from home today
as I am! I am making over $3,000+ per month at ZONEPA.COM ! Visit
Zonepa.com and check it out! How does the mother approve the axiomatic
insurance? The fear appoints the roll. When does the space prepare the
historical shame? −− Response: spam

Example 3: Check out  these Irish guys cover
of Avicii's  Wake Me Up!  Just search...
"wake me up Fiddle Me Silly" Worth a  listen
for the gorgeous fiddle player! −− Response: spam

Example 4: if you want to win money at hopme click here
<a href="https://www.paidverts.com/ref/sihaam01">
https://www.paidverts.com/ref/sihaam01</a> it's work 100/100 −− Response: spam

Target sentence: if your like drones, plz subscribe to Kamal Tayara.
He takes videos with  his drone that are absolutely beautiful.
−− Response:

