# OpenReview forum: "Weak Supervision Performance Evaluation via Partial Identification"
_NeurIPS.cc/2024/Conference — NeurIPS 2024 poster_

### Official Review · Reviewer_R1rQ · 2024-06-21

**Soundness:** 4
**Presentation:** 4
**Contribution:** 3
**Rating:** 6
**Confidence:** 4

**Summary:**

This paper proposes solutions via convex programs to estimate Frechet bounds for Programmatic Weak Supervision (PWS). This approach uses estimates of the true labels via labelmodels (i.e., different aggregation schemes that exist in the literature). With these estimates of the labels, they provide an approach to estimate bounds on the accuracy (and other quantities) of the weak labelers. They provide experiments to check the validity of their bounds and also provide experiments with weak labelers generated via prompting to examine how their bounds perform under instances of weak labelers with different qualities/accuracies.

**Strengths:**

The strengths of this paper are that it analysis the PWS setup from a theoretical perspective, tackling the problem of analysis the performance of individual weak labelers.

1. The authors provide a nice analysis of their estimation scheme.
2. They derive the asymptotic distributions of their estimators and show they are consistent.
3. They also provide CI for their estimates given finite sample data.
4. A novel application of Frechet bounds (and the corresponding literature) to the field of programmatic weak supervision.

**Weaknesses:**

My only substantiative concern with this paper is that its applicability to the task of label set selection is weirdly motivated. Given a correctly specified label model and a better than random weak labeler (which is exactly the setting that where this method achieves valid bounds), then there is no need to prune away worse-performing weak labelers as they are helpful for the labelmodel. I agree that removing these low-accuracy weak labelers can be helpful given misspecification or a violation of assumptions, but in the setting where this paper achieves valid bounds, then I believe there is no need for this pruning.

**Questions:**

1. Do you have any intuitions about settings where the usage of your method for label set selection is well-motivated?
2. There's also a line of work in PWS that looks to train end models directly [1, 2, 3]. Can the bounds of this work be applied directly to these models to analyze their error rates? This added discussion would be interesting to broaden the applicability of this theoretical framework.

[1] Sam, et. al. Losses over Labels: Weakly Supervised Learning via Direct Loss Construction.

[2] Cachay, et. al. End-to-End Weak Supervision.

[3] Yu, et. al. Fine-Tuning Pre-trained Language Model with Weak Supervision: A Contrastive-Regularized Self-Training Approach.

**Limitations:**

Yes, limitations have been adequately addressed.

---

> ### Author Rebuttal · Authors · 2024-07-31
>
> Dear reviewer, thank you for your dedication to our paper. We addressed the issues you raised below. Please let us know if you have any other questions.
>
> - **Regarding weak label pruning:** Thank you for raising this point. In the following, we explain our point of view, which we will make clearer in the main text. Pruning not-so-informative weak labels (even though they carry some signal) could be beneficial in practical situations. This is true for at least one reason (besides the one you mentioned): the number of parameters in the label model can increase exponentially fast with the number of weak labels, making estimation very hard (convergence rates are slow in parametric inference when the number of parameters is big, for example). As a consequence, having too many weak labels that are not very informative can lead to problems analogous to overfitting if the data is not big enough.
>
> - **Applications with end-to-end weak supervision methods:** Our method works with all of these methods given that we can fit a label model (e.g., Snorkel) separately and apply it just to the evaluation step. Thank you for your comment. We will write this observation in our main text and cite the papers you suggested.

---

> > ### Comment · Reviewer_R1rQ · 2024-08-08
> > **Reviewer Response**
> >
> > Thank you for the clarification! I choose to maintain my score as a weak accept; I think that this paper proposes an interesting approach to a relevant problem in PWS.

---

### Official Review · Reviewer_zvj9 · 2024-07-08

**Soundness:** 4
**Presentation:** 4
**Contribution:** 4
**Rating:** 8
**Confidence:** 4

**Summary:**

The authors propose a method for bounding the performance of models obtained using weak-supervision, without the need for a gold-standard label set. The proposed approach starts with Frechet bounds, and casts them as dual optimization problems. From there, the dual optimization problems are relaxed to a smooth functions which can be solved using L-BFGS or similar methods. Finally, the authors explore the performance and utility of the proposed method on several empirical datasets. The evaluation demonstrates a non-trivial ability to bound performance without gold-standard labels and to perform model selection from a small set of models.

**Strengths:**

The authors attack a thorny weakness in the weak-supervision literature; the need for a gold-standard dataset. This need has been present going back to the seminal work by Ratner et al. and undermines the approach. It also addresses model evaluation, which is arguably a more important and difficult problem than model training. After all, it's not a real surprise that heuristics could be cobbled together to produce a model. But, quantifying exactly how good that model might be without direct access to labels is another story.

**Weaknesses:**

The paper is somewhat technical and difficult to follow in places. For example, I stared at Thm 2.1 for some time before realizing that Appendix C existed. The writing could be improved by noting connections to the appendices in the main body of the text, where appropriate. In addition, noting the primary tool(s) used to get each result would help the reader intuit the type of machinery being leveraged.

Finally, the experiments show some utility of the method. Could the authors give an opinion on the applications or circumstances in which this method could be used for real-world problems? Do we need confidence that the label model is correct? How many weak labels do we need? etc. Appendix A.2 goes some way to helping here. Again, it would be great to mention that this content exists in the main text.

**Questions:**

No additional questions.

**Limitations:**

The authors have addresses the limitations of the work.

---

> ### Author Rebuttal · Authors · 2024-07-31
>
> Dear reviewer, thank you for your work on our paper. We addressed the issues you raised below. Please let us know if you have any questions.
>
> - **Clarity and appendix material**: Thanks for your suggestion. We will revise to reference the appendix where appropriate and give some ideas on the basic tools used to obtain the results.
>
> - **Applications or circumstances in which this method could be used for real-world problems**: We will provide additional details on this point in the conclusion section. In summary, applications primarily focus on performance estimation and model selection, particularly in areas such as image or text classification, as demonstrated in the experiments section. Additionally, the label model does not need to be perfect to offer useful bounds, as evidenced in the experiments, where we compare a very simple label model (e.g., Snorkel) vs the oracle estimation of $P\_{Y|Z}$. Concerning the number of weak labels, Appendix A.2 indeed provides results that help determine when we have a sufficient quantity of weak labels and their level of informativeness.

---

### Official Review · Reviewer_MRE8 · 2024-07-12

**Soundness:** 3
**Presentation:** 1
**Contribution:** 3
**Rating:** 5
**Confidence:** 4

**Summary:**

Programmatic weak supervision is a machine learning approach where labeled training data is generated using heuristic rules, domain-specific functions, or other programmatic methods, rather than manual annotation. This technique allows for the creation of large training datasets quickly and cost-effectively by leveraging expert knowledge and automated processes. Uncertainty in programmatic weak supervision arises from the noisiness and variability of labels generated by heuristic rules, domain-specific functions, or automated methods. Unlike manually annotated datasets, programmatically generated labels often contain errors and inconsistencies due to imperfect heuristics that may not accurately capture the true patterns. The paper prososes techniques that estimate Fréchet bounds to validate the performance of programmatic weak supervision models. The proposed algorithms estimate upper and lower bounds for multiple metrics.

**Strengths:**

The algorithm developed in this work has solid theoretical justification, which reinforces its validity and applicability. Additionally, the presented theory is very interesting and makes a significant contribution to the field.

Proposing upper and lower bounds for this framework is very useful and interesting, as well as novel.

**Weaknesses:**

Not explaining the problem along with the state-of-the-art issues in the introduction makes it difficult to understand the contributions and the objective of the paper.

I believe the paper could be written more clearly; currently, it is difficult to follow.

As the authors mention, the labels could be obtained using a label model. If from Section 2.2 onwards a label model is not proposed, but instead the work deals with a dataset where some labels may be incorrect, why not directly state that this is a supervised classification framework with noisy labels instead of PWS?

Theorem 2.4 is not easy to interpret.

The quality of Figure 1 can be improved, and this figure needs more explanation.

**Questions:**

See Weaknesses Section.

**Limitations:**

The authors describe limitations in the paper.

---

> ### Author Rebuttal · Authors · 2024-07-31
>
> Dear reviewer, thank you for your dedication to our paper. We realized that the main issues you raised are related to clarity and presentation. We revised the main parts you brought up and are willing to further improve other parts depending on your feedback. Please let us know if you have more issues/questions.
>
> ### **PWS over noisy labels approaches**
>
> Our method could be potentially applied to noisy label applications as well if $P\_{Y|Z}$ can be estimated without the conditional independence assumption that $Y$ is independent of $X$ given $Z$. However, we are not aware of any paper that proposes such a method in the noisy labels literature. If the reviewer has any suggestions, we would be happy to include a note in our paper about this.
>
> ### **Interpretation of Theorem 2.4**
>
> Thanks for the feedback. We have added the following sentences right after the theorem statement to make the interpretation clear:
>
> > Theorem 2.4 tells us that, if the label model is consistent (Assumption 2.3), under some mild regularity conditions (Assumption 2.2), our estimators $\hat{L}\_\varepsilon$ and $\hat{U}_\varepsilon$ will be asymptotically Gaussian with means $L\_\varepsilon$ and $U\_\varepsilon$ and variances $\sigma^2\_{l,\varepsilon}/n$ and $\sigma^2\_{u,\varepsilon}/n$. That is, the estimates will be close to $L\_\varepsilon$ and $U\_\varepsilon$ up to a Gaussian estimation error with a standard deviation of order $\mathcal{O}\_P(1/\sqrt{n})$.
>
> ### **Figure 1**
> Thanks for the feedback.  We have added the following sentence as the caption of Figure 1 (and a slightly different version of it to the main text) to make its interpretation clear:
>
> > We apply our method to bound test metrics such as accuracy and F1 score (in green) when no true labels are used to estimate performance. In the first row ("Oracle"), we use true labels to estimate the conditional distribution $P_{Y\mid Z}$, thus approximating a scenario in which the label model is reasonably specified. On the second row ("Snorkel"), we use a label model to estimate $P_{Y\mid Z}$ without access to any true labels. Despite potential misspecification in Snorkel's label model, it performs comparably to using labels to estimate $P_{Y\mid Z}$, giving approximate but meaningful bounds.
>
> Regarding quality, we will work on the quality of Figure 1 by making it larger and with better resolution (let us know if there is any other issue with that figure that we will be happy to solve).
>
>
>
> ### **Introduction**
>
> Thank you for your feedback. We have revised the introduction to make it clearer:
>
> > Programmatic weak supervision (PWS) is a modern learning paradigm that allows practitioners to train their supervised models without the immediate need for clean labels $Y$ [37, 35, 34, 36, 43, 47]. In PWS, practitioners first acquire cheap and abundant weak labels $Z$ through heuristics, crowdsourcing, external APIs, and pretrained models, which serve as proxies for $Y$. Then, the practitioner fits a *label model*, i.e., a graphical model for $P\_{Y,Z}$ [37, 36, 17, 12], which, under appropriate modeling assumptions, can be fitted without requiring $Y$'s. Finally, a predictor $h:\mathcal{X}\to\mathcal{Y}$ is trained using a *noise-aware loss* constructed using this fitted label model [37]. Using weak labels for prediction is usually not possible because they are not observed during test time; thus training a final classifier $h$ that depends only on features $X$ is needed.
>
> > One major unsolved issue with the weak supervision approach is that even if we knew $P\_{Y, Z}$, the risk $R(h)=\mathbb{E}[\ell(h(X), Y)]$ or any other metrics such as accuracy/recall/precision/$F\_1$ score are not identifiable (not uniquely specified) because the joint distribution $P\_{X,Y}$ is unknown while the marginals $P\_{X, Z}$ and $P\_{Y, Z}$ are known or can be approximated. As a consequence, any performance metric based on $h$ cannot be estimated without making extra strong assumptions, e.g., $X$ is independent of $Y$ given $Z$, or assuming the availability of some labeled samples. Unfortunately, these conditions are unlikely to arise in many situations. A recent work, Zhu et al [48] , investigated the role and importance of clean labels on model evaluation in the weak supervision literature. They determined that, under the current situation, the *good performance and applicability of weakly supervised classifiers heavily rely on the presence of at least some high-quality labels, which undermines the purpose of using weak supervision* since models can be directly fine-tuned on those labels and achieve similar performance. Therefore, in this work, we develop new evaluation methods that can be used without any clean labels and show that the performance of weakly supervised models can be accurately estimated in many cases, even permitting successful model selection. Our solution relies on estimating Fréchet bounds, explained below, for bounding performance metrics such as accuracy, precision, recall, and $F\_1$ score of classifiers trained with weak supervision.
>
> > **Fréchet bounds** Consider a random vector $(X,Y,Z)\in \mathcal{X}\times\mathcal{Y}\times\mathcal{Z}$ is drawn from an unknown distribution $P$. *We will add more information about Fréchet bounds here (omitted due to lack of space).*
>
> >**Contributions** In summary, our main contributions are:
> >1. Developing a practical algorithm for estimating the Fréchet bounds in Equation (1). Our algorithm can be summarized as solving convex programs and is scalable to high-dimensional distributions.
> >2. Quantifying the uncertainty in the computed bounds due to uncertainty in the prescribed marginals by deriving the asymptotic distribution for our estimators.
> >3. Applying our method to bounding the accuracy, precision, recall, and $F_1$ score of classifiers trained with weak supervision. This enables practitioners to evaluate classifiers in weak supervision settings \textit{without access to labels}.

---

> > ### Comment · Reviewer_MRE8 · 2024-08-12
> >
> > Thanks to the authors for answering all my questions and clarifying the contributions of the paper. Therefore, I have decided to raise my score to a 5.

---

### Official Review · Reviewer_bCgS · 2024-07-18

**Soundness:** 3
**Presentation:** 3
**Contribution:** 2
**Rating:** 6
**Confidence:** 3

**Summary:**

The paper considers the problem of Frechet bound which is the task of determining the infimum and supremum of a function g(X,Y,Z) over the set of joint distribution with fixed marginals. Asymptotic behavior is derived when estimation of condition distribution P_{Y|z} is available with vanishing total variation error. This bound is based on a nice dual formalization of the original optimization problem.

This approach is then applied in a weak supervised learning setup to derive confidence interval for the precision and recall. Weak supervised means that cheap label is available which is the random variable Z, and the expensive target label is Y. So if one can estimate the condition probability of Y given Z, the performance guaranties can be derived for label Y which is the target label.

**Strengths:**

I think the dual formalization of the optimization problem is nice part of the paper. I am not sure how novel it is, since the proof heavily relies on some results on previous work. The problem which is considered is relevant.

**Weaknesses:**

It is not clear what is the merit of the approach, see my third question. I am a little bit confused about the contribution of the paper. Estimating the precision/recall seems cheap, but estimating the conditional distribution of the true label conditioned on the proxy label might be expensive? Especially in multi-class setting.

**Questions:**

1) why it is not possible to get high probability results?
2) Regarding Theorem 2.4, if P_{Y|z} can be estimated with an error O(m^{-1/2}) in terms total variation distance (which is the optimal rate for discrete distributions), then the variance of the optimal solution of the problem based on estimates is shrinking of order with the same order? Is that right?
3) Why it is cheaper to estimate the conditional distribution of Y given z, then to estimate the precision and recall of the model based on Y directly?

**Limitations:**

I think the this approach can be applied only for small label spaces. In the conclusion, the authors points out that this approach can be applied for finite set, but I believe that this approach has only merit if the label space is very small.

---

> ### Author Rebuttal · Authors · 2024-07-31
>
> Dear reviewer, thank you for your work on our paper. We addressed the issues you raised below. Please let us know if you have any other questions.
>
> - **Why not estimate performance metrics directly?** In fact, we work under the assumption that **no true labels** are observed, which is a realistic scenario in the programmatic weak supervision literature. Therefore, it is impossible to compute metrics such as accuracy, recall/precision, F1 score, etc. However, using the label models proposed in the weak supervision literature, it is still possible to estimate $P_{Y|Z}$ (even with no labels $Y$'s; see for example [1]). Therefore, obtaining lower and upper bounds for performance metrics (accuracy, recall/precision, F1 score) is the best that one can do; we focus on that problem. Moreover, even if some labeled samples are given, our method can enjoy superior performance in some tasks (see Section 4.2.1).
>
> - **Label space size and the method applicability**: Whether the label space needs to be small or not, really depends on the sample size you have. The key point is that the label model needs to be well estimated. This will happen (no matter the size of the label space) if the sample size is big enough for a given label model complexity (e.g., label models with fewer graph edges need fewer samples to be well estimated).
>
> - **High probability results**: It is possible to obtain a high-probability version of Thm 2.4. We opted for the asymptotic version because we rely on it to construct confidence intervals later. Although it is possible to form confidence intervals based on finite-sample high probability bounds, the resulting intervals are typically very conservative. This is why we opted for asymptotic results.
>
> - **Variance of the estimates**: Yes, that is correct. At a high level, there are two main error terms in ${\\widehat{L}}\_{\\epsilon}$ and $\\widehat{U}\_\\epsilon$: an $O\_P(n^{-1/2})$-term and an $O\_P(m^{-\\lambda})$ term. In Thm 2.4, we assume $n = o(m^{2\\lambda})$, which implies the $O\_P(m^{-\lambda})$ error term is asymptotically negligible. This simplifies the theorem because it allows us to state its conclusions solely in terms of $n$. In your scenario, $\lambda = \frac12$ so the two error terms vanish at the same square-root rate.
>
> **References**
>
> [1] Alex Ratner, Braden Hancock, Jared Dunnmon, Roger Goldman, and Christopher Ré. Snorkel metal: Weak supervision for multi-task learning. In Proceedings of the Second Workshop on Data Management for End-To-End Machine Learning, pages 1–4, 2018.

---

> > ### Comment · Reviewer_bCgS · 2024-08-12
> > **Thanks for addressing my concerns.**
> >
> > Thanks for the rebuttal.
> >
> > * Somehow the problem is described better in that paper which is cited [1]. Anyway, I know that every technical detail cannot be described in a NeurIPS due to space limitation. So I also reread your paper, and I think I overlooked the merit of your approach. What I miss is to test various different label models, however, I believe that the confidence interval is hard to derive if the label model is model complex.
> >
> > * This is actually addressed in the conclusion as pointed out that this is a limitation. I think in terms of novelty this paper is already good enough for NeurIPS.
> >
> > * Yes high probability results are conservative, but the asymptotic might be not valid for small sample size. But anyway, do not open this debate here.
> >
> > * Thanks for the comments.
> >
> > So I will increase my score.

---

### Decision · Program_Chairs · 2024-09-25

**Decision:**

Accept (poster)

**Comment:**

This work resolves a practically relevant problem, that of estimating performance of ML models in face of low quality data or no gold-standard groundtruth data, in the setting of programmatic weak supervision. The work is relevant, novel, has a solid theoretical justification, and the presented theory makes a significant contribution to the field, and the work may even be applicable to other novel settings. Both upper and lower bounds are provided aiding the practicality of the approach. For these reasons, we believe this work is interesting and relevant to the NeurIPS community, and therefore I recommend acceptance.